# Towards Efficient Chain-of-Thought Reasoning via Adaptive-Budgeting based Policy Optimization

## Abstract

Recently, *Chain-of-Thought* (CoT) reasoning has become a key problem-solving capability for advanced *large language models* (LLMs) to address difficult tasks such as the mathematical ones. However, balancing the efficiency and performance of long CoTs still remains an intractable challenge. In this paper, we observe that assigning adaptive token budgets for different examples during training is an viable way to tackle with the above issue. Motivated by this, we propose a novel reinforcement learning scheme, termed *Adaptive-Budgeting based Policy Optimization* (ABPO). Based on the popular GRPO, our ABPO redefines the RL training as an adaptive curriculum learning process, where example pools are curated to categorize training examples into three types, namely the mastered, learning and hard ones, respectively. As the training progresses, ABPO will adaptively schedule the examples with proper length budgets, and the example pools will alse be dynamically updated based on the model status. In this way, we can assign adaptive token lengths for different examples during RL training, achieving a good balance between efficiency and performance of CoTs. To validate ABPO, we apply it to three representative LLMs, and conduct extensive experiments on a bunch of CoT reasoning benchmarks. The experimental results not only show the substantial efficiency improvements with minimal performance loss, *e.g.*, reducing token length by 78.3% while improving 2.0% performance of DeepSeek-R1-Distill-Qwen-1.5B on average, but also show our obvious advantages over the compared methods, *e.g.*, reducing 59.4% length and increasing 8.3% performance on average than HAPO, respectively. Our code is anonymously released at https://anonymous.4open.science/r/AnonymizeABPO-5380/

## 1 Introduction

Recently, *test time scaling* Muennighoff et al. (2025) has become the *de facto* paradigm of advanced *large language models* (LLMs) Grattafiori et al. (2024); Yang et al. (2024); Bai et al. (2023); Achiam et al. (2023). Following this paradigm, advanced LLMs DeepSeek-AI (2025); Team (2025) are trained in reinforced or supervised manners to generate long *chain-of-thoughts* (CoTs), thereby handling difficult tasks, such as mathematical problem solving of America (2024); Gao et al. (2024) and logical reasoning Xie et al. (2025). Despite the great success, "*over-thinking*" Chen et al. (2025) becomes another notable issue for these LLMs. For instance, Qwen-QwQ Team (2025) will generate CoTs of more than $2k$ tokens to answer a simple question, *e.g.*, *"The product of integers 240 and $k$ is a perfect cube. What is the smallest possible positive value of $k$?"* This case will become more prominent for smaller LLMs, such as DeepSeek-R1-Distill-Qwen-1.5B and DeepScaleR-1.5B-Preview, which strongly contradicts their goals about efficient applications.

To address this problem, recent efforts are devoted to the research of efficient CoT reasoning Muennighoff et al. (2025); Aggarwal & Welleck (2025); Ma et al. (2025); Luo et al. (2025a). Among these progresses, one plausible strategy is to imply length-related rewards to the RL training of LLMs Aggarwal & Welleck (2025); Yeo et al. (2025); Arora & Zanette (2025); Luo et al. (2025a). In particular, LLMs are assigned by a fixed-length token budget during its training. And when the output CoT exceeds the token budget, a negative reward will be returned to guide the RL optimization Aggarwal & Welleck (2025); Yeo et al. (2025).

In this way, LLMs are forced to learn to solve the tasks using limited-length CoTs.

However, this fixed-budget based solution still encounters several limitations. Above all, the example difficulties for LLMs vary greatly, even for the mathematical tasks. And fixing the token budget is hard to meet the trade-off between efficiency and performance. To explain, as shown in Fig.1, when the token budget is small, LLMs are prone to under-cutting the hard examples while over-supplying the easy ones. For instance, under a budget of $2k$ tokens, DeepSeek-R1-Distill-Qwen-1.5B can perform well on the simple tasks like MATH500 but receives obvious performance drops on the more difficult ones, *e.g.*,-25% on AIME of America (2024). Moreover, simply increasing the token budget will makes the efficiency benefit become limited, *e.g.*, $4k$ tokens, and the tuning cost also increases substantially . In this case, we argue that allocating appropriate token budgets to different examples based on the model status is the key to addressing this shortcoming.

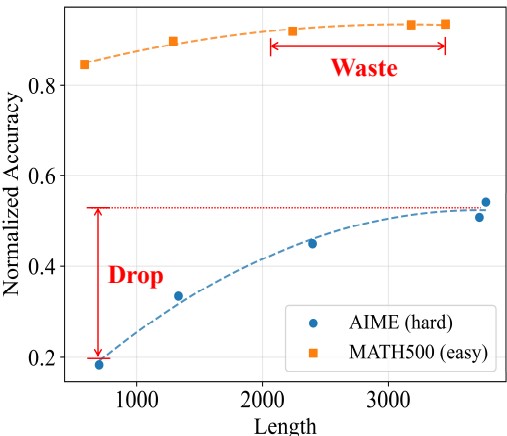

Figure 1: The performance–length curve of L1-Exact 1.5B Aggarwal & Welleck (2025) on two datasets. For easy questions, CoT will result in an obvious wast of computation. In contrast, for the hard ones, the limited token budgets of CoT will lead to obvious performance drop. Thus, assigning proper token budgets is the key to achieve the balance between efficiency and performance.

However, achieving the adaptive budgeting is notoriously challenging in existing RL training scheme. Although the example difficulty for LLMs can be measured via accuracy, entropy or prediction confidence Su & Cardie (2025); AlKhuzaey et al. (2024), the model's assessment of difficulty is a dynamic process that changes as training progresses, and how to assign proper token budgets is still intractable. Besides, the training scheduler also requires in-depth exploration. Gradually increasing the token budget is a natural solution, *e.g.*, *curriculum learning* Bengio et al. (2009), but it also causes the LLM to continuously traverse the simple examples, resulting additional issues of training efficiency and expenditure, expecially considering the expensive cost of the widely used GRPO training Shao et al. (2024). Therefore, how to effectively and dynamically adjust proper token budgets still remains an open problem.

To address these issues, we propose a novel training scheme termed *Adaptive-Budgeting based Policy Optimization* (ABPO), which extends the popular *Group Relative Policy Optimization* (GRPO) Shao et al. (2024) with adaptive and budget-aware rewards in a dynamic training process. Concretely, ABPO will first let the LLM to quickly read training examples, and then categorize the examples into three pools, namely the *mastered*, the *learning* and the *hard* ones, respectively. In practice, ABPO starts from a low token budget, which is then gradually increased. During training, the hard examples are not directly used, but they will be adaptively updated to the *learning* pool based on the model status. Meanwhile, we also introduce an efficient review mechanism for the *mastered* examples, ensuring the knowledge retention and avoiding catastrophic. This dynamic scheduling can help to assign model-aware token budgets for different examples and avoid the redundant training of examples. In this case, ABPO can achieve a good trade-off among CoT efficiency and performance as well as training expenditure.

To validate ABPO, we apply it to a set of advanced LLMs, including DeepSeek-R1-Distill-Qwen-1.5B, 7B DeepSeek-AI (2025) and DeepScaleR-1.5B-Preview Luo et al. (2025b), on four representative reasoning benchmarks of varying difficulties, which are MATH500 Hendrycks et al. (2021), AIME of America (2024), AMC AMC (2025) and OlympiadBench He et al. (2024). The experiment results show that ABPO can significantly reduce the token cost of CoT reasoning while maintaining competitive performance across various benchmarks. For instance, ABPO reduces the average CoT length of DeepSeek-R1-Distill-Qwen-1.5B by 78.3% while improving 2.0% of performance on average. Compared with prior CoT optimization methods, our ABPO can also obtain better CoT efficiency while achieving overall performance improvement on all benchmarks, *e.g.*, 2,217 *v.s.*, 5,466 length and 60.2 *v.s.* 55.6 accuracy compared to HAPO Huang et al. (2025). These results well confirm the effectiveness of our the motivation and designs of ABPO towards efficient CoT.

Figure 2: Illustration of *Adaptive Budgeting based Policy Optimization* (ABPO). During training, ABPO evaluates the difficulty of the training samples and groups them into three pools, *i.e.*, the mastered, learning and hard ones, respectively. As the training progresses, ABPO will adaptively allocate proper token budgets to the learning examples, and also actively review the mastered ones to avoid knowledge forgetting. In addition, these examples pools will be dynamically updated based on the LLM's status during training, thereby balancing efficiency, performance and training costs.

Overall, our contributions are three-fold:

- We show the limitation of using fixed-budget CoT optimization schemes in terms of balancing CoT efficiency and performance, and also identify the key challenges of implementing adaptive budget rewards in existing RL training.

- We propose a novel training scheme termed *Adaptive-Budgeting based Policy Optimization* (ABPO), which can adaptively assign token budgets and schedule the RL training.

- On a set of LLMs and benchmarks, ABPO shows better capability of balancing CoT efficiency and performance than the compared methods.

## 2 RELATED WORK

*Chain-of-Thought* (CoT) refers to generating step-wise reasoning traces before answering the given question Wei et al. (2022), which has become a critical capability of advanced LLMs DeepSeek-AI (2025); Luo et al. (2025b); Team (2025); Team et al. (2025). However, these reasoning LLMs often produce excessively long CoT that commonly exhibit too many branching, verification, and backtracking. To address this issue, several works have recently been proposed for efficient CoT Aggarwal & Welleck (2025); Hou et al. (2025); Yeo et al. (2025); Hammoud et al. (2025). Specifically, one popular solution is using budget-based rewards in RL training Aggarwal & Welleck (2025); Yeo et al. (2025); Hou et al. (2025); Hammoud et al. (2025). For example, L1 Aggarwal & Welleck (2025) introduces a penalty term to enforce length budgets specified in the prompt. ThinkPrune Hou et al. (2025) introduces a uniform budget limit for all training samples. Curriculum GRPO Hammoud et al. (2025) gradually reduce the number of tokens a LLM is allowed to use, which however ignores the differences among questions. However, this static method often hard to meet the trade-off between efficiency and performance. In this case, recent works Luo et al. (2025a); Huang et al. (2025); Arora & Zanette (2025); Su & Cardie (2025); Yi et al. (2025) start exploring dynamic-length rewards for LLMs. O1-Pruner Luo et al. (2025a) and Training Efficiently Arora & Zanette (2025) apply a length penalty based on the within-group comparison. ShorterBetter Yi et al. (2025) ties its reward function to the length of the shortest and correct response for each sample. Although these methods incorporate dynamic reward signals, they still need to repeatedly train on the mastered samples, which significantly reduces training efficiency. In contrast, our approach considers the budget and training schedule for different samples, achieving a balance between performance and efficiency.

## 3 METHOD

### 3.1 PRELIMINARY

*Chain-of-thought* (CoT) Wei et al. (2022) is a key capability of advanced LLMs in terms of long and complex task solving. The inference process of an LLM with CoT can be defined by

$$\hat{y}_i = LLM(o_i, q_i), \quad where \ o_i = LLM(q_i). \tag{1}$$

Here $q_i$ denotes the input question, $o_i$ is the generated reasoning trace, *i.e.*, CoT, and $\hat{y}_i$ is the predicted answer. Under the popular *Group Relative Policy Optimization* (GPRO) Shao et al. (2024), its objective can be defined by

$$\mathcal{J}_{GRPO}(\theta) = \mathbb{E}[q \sim D, \{o_i\}_{i=1}^G \sim \pi_{\theta_{old}}(O|q)]$$

$$\frac{1}{G}\sum_{i=1}^G \frac{1}{|o_i|}\sum_{t=1}^{|o_i|}\left[\frac{\pi_\theta(o_{i,t}|q, o_{i,<t})}{\pi_{\theta_{old}}(o_{i,t}|q, o_{i,<t})}\hat{A}_{i,t} - \beta L_{KL}(\pi_\theta, \pi_{ref})\right], \tag{2}$$

$$where \ \hat{A}_{i,t} = \frac{\mathbb{I}[\hat{y}_i = y_i] - mean(R)}{std(R)}.$$

Here, $q$ denotes the sampled query from the dataset $D$, $o_i$ is the $i$-th generated output sampled from the old policy $\pi_{\theta_{old}}$, $\pi_\theta$ is the updated policy for optimization, and $\pi_{ref}$ is the reference model used for KL-regularization. $\hat{A}_{i,t}$ is the standardized reward advantage for token $t$ with $R$ denoting the set of rewards. $\beta$ controls the strength of the *KL-regularization* $L_{KL}(\pi_\theta, \pi_{ref})$. In terms of the default GPRO, the reward function does not consider the length of CoT outputs, so the training mainly focuses on the prediction accuracy, which makes the model prone to *over-thinking*. And this issue will become more obvious in the smaller distilled LLMs Cuadron et al. (2025).

In this case, some recent works Hou et al. (2025); Aggarwal & Welleck (2025); Yeo et al. (2025) implement length-related reward to RL training. When the output CoT exceeds the max number of tokens, the LLM will receive a negative reward to penalize its optimization. As discussed above, the length-aware reward can encourage the model to learn to answer the questions with limited length of CoTs. However, the fixed budget commonly used Aggarwal & Welleck (2025); Hou et al. (2025) will also make the model under-cutting the hard examples while over-supplying the easy ones. In this case, we aim to achieve dynamic RL training with adaptive token budget scheduling.

### 3.2 ADAPTIVE-BUDGETING BASED POLICY OPTIMIZATION

In this paper, we propose a novel training scheme to balance the efficiency and performance of LLM's CoT reasoning, termed *Adaptive-Budgeting based Policy Optimization* (ABPO), of which illustration is depicted in Fig. 2. Concretely, ABPO introduces adaptive token budgets to the training of GPRO, and then the objective can be defined by

$$\mathcal{J}_{ABPO}(\theta) = \mathbb{E}[q \sim D, \{o_i\}_{i=1}^G \sim \pi_{\theta_{old}}(O|q)]$$

$$\frac{1}{G}\sum_{i=1}^G \frac{1}{min(|o_i|, b(q_i))}\sum_{t=1}^{min(|o_i|, b(q_i))}\left[\frac{\pi_\theta(o_{i,t}|q, o_{i,<t})}{\pi_{\theta_{old}}(o_{i,t}|q, o_{i,<t})}\hat{A}_{i,t} - \beta L_{KL}(\pi_\theta, \pi_{ref})\right], \tag{3}$$

$$where \ \hat{A}_{i,t} = \frac{\mathbb{I}[\hat{y}_i = y_i \wedge |o_i| \le b(q_i)] - mean(R)}{std(R)}.$$

Here $b(q_i)$ is the dynamic token budget that LLM can use to answer the question $q_i$. And the budget-aware reward function can be defined by

$$r(o_i, \hat{y}_i) = \begin{cases} 1, & \text{if } \hat{y}_i = y_i \text{ and } |o_i| \le b(q_i), \\ 0, & \text{otherwise.} \end{cases} \tag{4}$$

where $o_i$ and $\hat{y}_i$ are the reasoning trace and the predicted answer for the input question $q_i$. Notably, $b(q_i)$ will be adaptively adjusted according to the difficulties of examples to the LLM.

Before training, ABPO first considers that all examples are unknown (hard) to the model, and evaluates their accuracies for the current model $\pi_\theta$. In practice, we will first generate $n$ CoT traces $O_i$ for each question $q_i$:

$$O_i = \{(o_{i,j}, \hat{y}_{i,j}) \mid (o_{i,j}, \hat{y}_{i,j}) = \pi_{\theta,j}(q_i)\}, \tag{5}$$

where $\pi_{\theta,j}$ represents the $j$-th sampling based on the policy model $\pi_\theta$. Notably, these traces can also be used as $\{o_i\}_{i=1}^G$ in Eq.3, avoiding the waste of computation. Then, the average accuracy of the model $\pi_{\theta_{old}}$ for the question $q_i$ under the length budget $b(q_i)$ can be obtained by

$$Acc(q_i) = \frac{1}{n} \sum_{j=1}^{n} \mathbb{I}[\hat{y}_{ij} = y_i \ \wedge \ |o_{i,j}| \leq b(q_i)], \tag{6}$$

where $\mathbb{I}[\cdot]$ is an indicator function. Initially, all samples are given the same budget $b(q_i) = t_0$. Via Eq. 6, we can categorize these examples into three pools, defined by

$$\mathcal{S}_{\text{mastered}} = \{q_i \mid Acc(q_i) \geq \tau_1\}, \tag{7}$$
$$\mathcal{S}_{\text{learning}} = \{q_i \mid \tau_1 > Acc(q_i) > \tau_0\}, \tag{8}$$
$$\mathcal{S}_{\text{hard}} = \{q_i \mid \tau_0 \geq Acc(q_i)\}. \tag{9}$$

Here, $\tau_0$ and the $\tau_1$ are the lower and upper thresholds, respectively. Thus, the mastered pool $\mathcal{S}_{\text{mastered}}$ contains examples that have been solved under existing token budget $b(q_i)$. The learning pool $\mathcal{S}_{\text{learning}}$ have samples with a certain probability of being learned. In contrast, the hard pool $\mathcal{S}_{\text{hard}}$ contains difficult examples that are not suitable to learn by LLMs, and they will not be used.

During ABPO training, the examples used are mainly composed of the *learning* ones, but we will still review a certain number of the *mastered* examples to avoid knowledge forgetting, with a review rate denoted by $\alpha$. After optimizing the LLM with the current batch, we calculate their accuracies according to Eq.6. Based on the accuracies, we can update the examples pools $S_{learning}$ and $S_{mastered}$. When the number of $\mathcal{S}_{\text{learning}}$ is reduced to a certain proportion $\lambda$, we will increase the length budgets $b(q_i)$ for the samples in $\mathcal{S}_{\text{hard}}$, and then start a new difficulty evaluation for these examples, which will accordingly update the pools of both hard and learning.

Specifically, the new budget for the questions $q_i \in \mathcal{S}_{\text{hard}}$ can be calculated by

$$b(q_i) = t_0 + (k-1) \cdot d, \quad where \ q_i \in \mathcal{S}_{\text{hard}}. \tag{10}$$

Here $k$ is the number of the current iteration and $d$ denotes the increased length budget. Then we evaluate the accuracy of samples from $\mathcal{S}_{\text{hard}}$ under the new budget and move some examples into $\mathcal{S}_{\text{learning}}$ according to Eq.8 and Eq.9. In this way, easy examples will be added to the learning and mastered pools, which cost only a small budget, and the more difficult examples will receive a larger budget. Due to page limit, the detailed algorithm of ABPO is given in Appendix A.2.

## 4 EXPERIMENTS

### 4.1 DATASET AND METRIC

The experiments are conducted on four widely-used benchmarks, including MATH500 Hendrycks et al. (2021), AIME of America (2024), AMC AMC (2025), and Olympiad-Bench He et al. (2024). Among these benchmarks, MATH500 is often regarded as the easiest one, which is a subset of MATH Hendrycks et al. (2021). AMC and AIME are moderately difficult, which has examples from mathematics competitions. In particular, AIME is regarded as the most difficult one, of which examples are from American Invitational Mathematics Examination. Following the previous works Arora & Zanette (2025), the accuracy of an example is defined by the average performance across $K$ runs. During the evaluation, we set $K = 3$ for MATH500, $K = 5$ for Olympiad-Bench and $K = 10$ for both AIME and AMC. Additionally, we use the *Accuracy–Efficiency Score* (**AES**) Luo et al. (2025a) to evaluate the trade-off between accuracy and computation overhead. A higher AES indicates a better balance between accuracy and efficiency.

### 4.2 IMPLEMENTATION DETAILS

We train ABPO on three base models, *i.e.*, DeepSeek-R1-Distill-Qwen-1.5B, DeepSeek-R1-Distill-Qwen-7B DeepSeek-AI (2025), and DeepScaleR-1.5B-Preview Luo et al. (2025b). The main experiments are conducted on $2.4k$ training samples from the *AIME-AMC* subset of the Prime dataset Cui et al. (2025). To further validate the effectiveness of ABPO, we additionally train DeepSeek-R1-Distill-Qwen-1.5B on 32k samples from the AM dataset Zhao et al. (2025). For training steps,

Table 1: Comparison with existing efficient CoT optimization methods on three LLMs for four mathematical benchmarks. "*Acc.*", "*Len.*" and "*AES*" denote accuracy (higher is better), length (lower is better) and Accuracy–Efficiency Score (higher is better), respectively. The best and second best results are marked in **bold** and underline, respectively.

| | Train Data | MATH500 | | | AMC | | | Olympiad | | | AIME | | | **Average** | | |
|---|---|---|---|---|---|---|---|---|---|---|---|---|---|---|---|---|
| | | Acc. ↑ | Len. ↓ | AES ↑ | Acc. ↑ | Len. ↓ | AES ↑ | Acc. ↑ | Len. ↓ | AES ↑ | Acc. ↑ | Len. ↓ | AES ↑ | Acc. ↑ | Len. ↓ | AES ↑ |
| *Base model: DeepSeek-R1-Distill-Qwen-1.5B* | | | | | | | | | | | | | | | | |
| Baseline | – | 83.3 | 5278 | – | 70.3 | 9314 | – | 50.3 | 10820 | – | 32.0 | 15476 | – | 59.0 | 10222 | – |
| ShorterBetter | 40k | 68.9 | **864** | -0.03 | 48.3 | **1194** | -0.69 | 36.2 | **2072** | -0.60 | 19.7 | 4286 | -1.20 | 43.2 | **2104** | -0.54 |
| Training Efficient | 3.2k | 81.9 | 2536 | 0.44 | 70.3 | 5183 | 0.44 | 49.2 | 6818 | 0.27 | 29.0 | 10552 | -0.15 | 57.6 | 6272 | 0.27 |
| HAPO | 2k | 81.5 | 2633 | 0.39 | 69.0 | 4507 | 0.43 | 49.2 | 5458 | 0.38 | 22.7 | 9268 | -1.06 | 55.6 | 5466 | 0.18 |
| **ABPO** (Ours) | 2.4k | 80.9 | 1647 | 0.54 | 71.3 | 2537 | 0.77 | 49.7 | 3311 | 0.63 | **29.3** | 6028 | 0.19 | 57.8 | 3371 | 0.57 |
| **ABPO** (Ours) | 32k | **85.2** | 1434 | **0.80** | **74.3** | 2068 | **0.95** | **52.7** | 2141 | **0.95** | 28.7 | **3223** | 0.28 | **60.2** | 2217 | **0.85** |
| *Base model: DeepSeek-R1-Distill-Qwen-7B* | | | | | | | | | | | | | | | | |
| Baseline | – | 93.2 | 4086 | – | 89 | 6484 | – | 66.1 | 8686 | – | 56.3 | 12837 | – | 71.9 | 8536 | – |
| ShorterBetter | 40k | 85.6 | **1233** | 0.29 | 82.3 | **2443** | 0.24 | 56.7 | **2990** | -0.06 | 49.0 | **5673** | -0.09 | 64.5 | 3910 | 0.03 |
| Training Efficient | 3.2k | **89.1** | 2121 | 0.26 | **87.5** | 4300 | 0.25 | 61.9 | 5626 | 0.04 | **52.3** | 9410 | -0.09 | 65.0 | 5112 | -0.08 |
| **ABPO** (Ours) | 2.4k | **92.2** | 2243 | **0.40** | 87.3 | 3511 | 0.36 | **63.5** | 4852 | 0.25 | 48.7 | 8493 | -0.34 | **66.0** | 4354 | **0.08** |
| *Base model: DeepScaleR-1.5B-Preview* | | | | | | | | | | | | | | | | |
| Baseline | – | 88.2 | 3155 | – | 81.3 | 5067 | – | 58.3 | 5718 | – | 40.0 | 9041 | – | 66.9 | 5745 | – |
| L1-Exact | 40k | 82.1 | 3189 | -0.35 | 71.0 | 3067 | -0.24 | 50.7 | 3196 | -0.22 | 22.3 | 3715 | -1.62 | 56.5 | 3292 | -0.35 |
| L1-Max | 40k | 84.5 | **1527** | 0.31 | 75.5 | **1768** | 0.30 | 51.1 | **1733** | 0.07 | 27.0 | **2290** | -0.88 | 59.5 | **1830** | 0.13 |
| HAPO | 2k | 81.9 | 2537 | -0.16 | 70.5 | 4119 | -0.47 | 53.6 | 3976 | -0.10 | 31.3 | 6974 | -0.86 | 59.3 | 4401 | -0.33 |
| ThinkPrune | 2.4k | 85.5 | 2029 | 0.21 | 78.5 | 3125 | 0.21 | **55.0** | 3532 | 0.10 | 34.7 | 5743 | -0.30 | 63.4 | 3607 | 0.11 |
| **ABPO** (Ours) | 2.4k | **85.6** | 1874 | 0.26 | **79.8** | 2845 | **0.35** | **55.0** | 3302 | 0.14 | **37.3** | 5736 | **0.03** | **64.4** | 3439 | **0.21** |

Table 2: Comparison between fixed-budget baselines and ABPO on DeepSeek-R1-Distill-Qwen-1.5B on AMC and AIME.

| | AMC | | | AIME | | | **Average** | | | Training Time (h) |
|---|---|---|---|---|---|---|---|---|---|---|
| | Acc. ↑ | Len.↓ | AES | Acc.↑ | Len.↓ | AES | Acc. ↑ | Len.↓ | AES | |
| Base | 70.3 | 9314 | – | 32.0 | 15476 | – | 51.1 | 12395 | – | – |
| Fixed-Budget 2000 | 66.3 | **2320** | 0.47 | 24.0 | **5576** | -0.61 | 45.1 | **3948** | 0.09 | **15.5** |
| Fixed-Budget 3000 | 68.8 | 2850 | 0.59 | 23.3 | 6309 | -0.76 | 46.0 | 4980 | 0.13 | 22.2 |
| Fixed-Budget 4000 | **71.8** | 3600 | 0.68 | 25.7 | 6376 | -0.40 | 48.7 | 4988 | 0.36 | 28.0 |
| **ABPO** (Ours) | 71.3 | 2537 | **0.77** | **29.3** | 6028 | 0.19 | 50.3 | 4283 | **0.57** | 20.3 |

DeepSeek-R1-Distill-Qwen-1.5B is trained for 350 steps under both the standard and extended-data settings, while DeepSeek-R1-Distill-Qwen-7B and DeepScaleR-1.5B-Preview are each trained for 50 steps. The compared methods are **ShorterBetter** Yi et al. (2025), **Training Efficient** Arora & Zanette (2025), **HAPO** Huang et al. (2025), **L1-Exact**, **L1-Max** Aggarwal & Welleck (2025) and **ThinkPrune** Hou et al. (2025). The detailed experimental settings are provided in Appendix A.3.

### 4.3 QUANTITATIVE ANALYSIS

**Comparison with existing methods** In Tab.1, we first compare the performance and efficiency of ABPO with existing CoT optimization methods on three representative LLMs. From this table, we can first observe that the default LLMs (*Baseline*) often requires excessive tokens to generate CoTs on all benchmarks, *e.g.*, 5,278 tokens on MATH500 by DeepSeek-R1-Distill-Qwen-1.5B, showing notable over-thinking issues. ABPO and the compared methods can well alleviate this problem. For instance, via taking the shortest reasoning length within a sampled group, ShorterBetter can reduce the CoT length of DeepSeek-R1-Distill-Qwen-1.5B to 864 tokens on MATH500. However, we can also see that the compared methods have obvious compromises in performance, *e.g.*, -38.4% and -29.1% by ShorterBetter and HAPO on the difficult AIME. In addition, it can be seen that the efficiency benefits of some methods become marginal on the difficult tasks in order to reduce performance loss. For instance, the SOTA method *Training Efficient* requires about $10k$ and $6.8k$ token budgets on AIME and Olympiad for DeepSeek-R1-Distill-Qwen-1.5B, respectively, resulting worse *Accuracy-Efficiency Score* (AES). In terms of fixed-budget methods, *e.g.*, ThinkPrune Hou et al. (2025), it can maintain the CoT length at a relatively short output, but it obviously drops the performance by 13.2% on the difficult AIME. In contrast, ABPO can achieve a good balance

Table 3: Ablation study of ABPO on DeepSeek-R1-Distill-Qwen-1.5B. We progressively add Curriculum Learning, Adaptive Budget and Example Review to ABPO. $\star$ denotes the full setting of ABPO.

| | AMC | | | AIME | | | Average | | |
|---|---|---|---|---|---|---|---|---|---|
| | Acc.↑ | Len.↓ | AES↑ | Acc.↑ | Len.↓ | AES↑ | Acc.↑ | Len.↓ | AES↑ |
| Base | 70.3 | 9314 | – | 32.0 | 15476 | – | 51.1 | 12395 | – |
| + Curriculum Learning | 68.0 | 2652 | 0.56 | 24.7 | 5250 | -0.48 | 46.3 | 3951 | 0.21 |
| + Adaptive Budget | 68.0 | **2232** | 0.60 | 26.3 | **5219** | -0.22 | 47.2 | **3726** | 0.31 |
| + Review$\star$ | **71.3** | 2537 | **0.77** | **29.3** | 6028 | **0.19** | **50.3** | 4283 | **0.57** |

Table 4: Comparison between ABPO and the alternative data scheduling strategies on DeepSeek-R1-Distill-Qwen-1.5B. $Mix(\cdot)$ denotes the mixing of different types of examples for sampling.

| | AMC | | | AIME | | | Average | | |
|---|---|---|---|---|---|---|---|---|---|
| | Acc.↑ | Len.↓ | AES↑ | Acc.↑ | Len.↓ | AES↑ | Acc.↑ | Len.↓ | AES↑ |
| Base | 70.3 | 9314 | – | 32.0 | 15476 | – | 51.1 | 12395 | – |
| $Mix(\mathcal{S}_{learning}, \mathcal{S}_{hard})$ | 62.8 | **1942** | 0.26 | 16.3 | **3954** | -1.70 | 39.5 | **2948** | -0.37 |
| $Mix(\mathcal{S}_{learning}, \mathcal{S}_{mastered})$ | 69.8 | 2768 | 0.67 | 28.7 | 6180 | 0.08 | 49.2 | 4474 | 0.45 |
| **ABPO** (Ours) | **71.3** | 2537 | **0.77** | **29.3** | 6028 | **0.19** | **50.3** | 4283 | **0.57** |

between efficiency and performance. When increasing the amount of training data, *i.e.*, $32k$, ABPO can help DeepSeek-R1-Distill-Qwen-1.5B reduce about 78.3% CoT length while achieving 2.0% performance gains on average. These results well confirm the effectiveness of our ABPO.

To better show our advantages, we compare ABPO with length-control approaches in Fig. 3 (a), *i.e.*, L1-Exact and L1-Max Aggarwal & Welleck (2025). We plot the *length–accuracy* of the DeepScaleR-1.5B-Preview under different budget settings. Specifically, L1-Max controls the maximum output length of the model through prompting, whereas L1-Exact enforces the model to generate outputs with an exact length budget specified in the prompt. We show the average performance on the four widely used becnhmarks, *i.e.*, AMC, MATH500, AIME and Olympiad. From this figure, we can first observe that the original LLM takes a very large number of tokens to achieve better performance. Specifically, the original LLM achieves only 3.9% performance improvement while consuming nearly twice the token overhead of our ABPO. We can also observe that both L1-Exact and L1-Max can reduce the length of CoTs. When extending the CoT lengths, the performance of these two methods slightly increases but eventually reaches the ceiling. Compared with these two methods, the proposed ABPO achieves better performance with the same budget. As shown in the figure, ABPO obviously occupies a more upper left position in the length-performance curve. Overall, these results show that the proposed ABPO method can achieve a better balance between efficiency and effect and significantly improve the LLMs' reasoning ability.

**Comparison with fixed-budget Baselines** In Tab. 2, we compare ABPO to the fixed-budget methods under the same experimental settings. For the fixed-budget baselines, we assign a fixed token budget to all training samples. All samples are trained together, and this setting can serve as a direct counterpart to our adaptive scheduling strategy. From Tab. 2, we can first observe that this fixed-budget baselines will impose a negative impact on difficult examples. Specifically, training with a budget of $3,000$ tokens on AIME reduces the output length by 59.2% but leads to 27.2% performance drop. In comparison, ABPO requires 8.6% less training time and reduces the output length by 61.0%, while resulting in an 8.4% performance drop. This performance is higher than the result under the fixed budget of $3,000$. Another observation is that for medium-difficulty examples, ABPO achieves a better balance between accuracy and length. Although its accuracy is close to the baseline of a higher fixed budget, the output length is much shorter. For instance, on the AMC dataset, ABPO's accuracy is only 0.6% lower than that of Fixed-budget $4,000$, but its length is reduced by 29.5%, while obtaining a 13.2% improvement in AES. Overall, these results demonstrate that ABPO can provide reasonable CoT budgets for examples of varying difficulties.

**Ablation study** In Tab. 3, we ablate the key designs of ABPO on DeepSeek-R1-Distill-Qwen-1.5B. In this table, "+ *Curriculum Learning*" denotes training with curriculum scheduling and training batches are constructed by the examples from $\mathcal{S}_{\text{learning}}$ with CoT length limited by 4,000 tokens. "+ *Adaptive Budget*" further introduces dynamic budget scheduling, where their initial budget is set

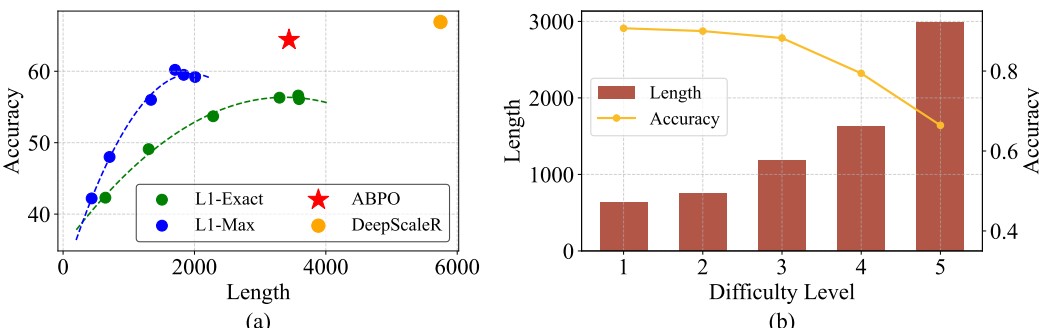

Figure 3: (a) Comparison between ABPO and efficient CoT optimization methods under different length budgets on DeepScaleR-1.5B-Preview. The accuracy is averaged over AMC, MATH500, AIME and Olympiad. (b) The average CoT length and accuracy of ABPO on examples of MATH500 at different difficulty levels. It can be seen that the proposed ABPO can generate CoT with different lengths according to the example difficulty.

to $t_0$=2,000 and gradually expanded with $d$=500 per iteration. Finally, "+ *Review*" corresponds to the full settings of ABPO, which additionally adds the review mechanism by mixing a fraction of the mastered samples. From Tab. 3, we can first observe that using curriculum learning can reduce the CoT length substantially. For instance, on AMC benchmark,"+ *Curriculum Learning*" reduces the reasoning trace by 71.5% with 3.3% performance drop. After adding adaptive budgeting, *i.e.*, "+ *Adaptive Budget*", the output length is further shortened while improving accuracy. For instance, the length of CoT is shorten by 76.0%, while keeping the same performance on AIME benchmark. With "+ *Review*", ABPO achieves the best trade-off, and it restores the accuracy close to the original LLMs while significantly reducing CoT token length. For instance, on AMC, the proposed ABPO method not only reduces the CoT length by 72.8%, but also improve the performance by 1.4%. These results well confirm the designs of the adaptive budget assignments and the example review.

In Tab. 4, we also examine the roles of three example pools in ABPO. We conduct two experiments with different sampling strategies while keeping all other configurations identical to ABPO. Here, "$Mix(\mathcal{S}_{learning}, \mathcal{S}_{hard})$" combing the examples of $\mathcal{S}_{learning}$ with $\mathcal{S}_{hard}$ for sampling, *i.e.*, not considering the difficulty of examples for training. "$Mix(\mathcal{S}_{learning}, \mathcal{S}_{mastered})$" denotes the mixing of $\mathcal{S}_{learning}$ and $\mathcal{S}_{mastered}$, *i.e.*, all examples are either under-learning or too hard for training. In contrast, ABPO will assess examples via three types while reviewing partial examples from $\mathcal{S}_{mastered}$, as described in Sec. 3.2. From these results, we can observe that "$Mix(\mathcal{S}_{learning}, \mathcal{S}_{hard})$" significantly reduces the CoT length of the LLM. For instance, it reduces CoT length by 74.4% on AIME. However, under this scheme, the longer CoTs for difficult examples always give the LLM a penalty signal, causing the LLM to mistakenly tend to output shorter CoT. On the other hand, we can observe that "$Mix(\mathcal{S}_{learning}, \mathcal{S}_{mastered})$" significantly reduces the CoT length while obtaining better performance. For example, "$Mix(\mathcal{S}_{learning}, \mathcal{S}_{mastered})$" reduces the inference length by 60.1% with 10.3% performance drop on AMC. To explain, after identifying the hard examples, this scheme can help the LLM achieve better optimizations during training. However, its performance is still limited due to the excessive use of easy examples, especially on the difficult ones. In contrast, via categorizing examples into three pools with a review design, ABPO can help the LLM better compensate the performance drops while keeping low CoT budgets. Overall, these ablation studies well confirm the designs of the example pools and the review operation in ABPO.

**Inference cost *w.r.t.* example difficulty.** In Fig. 3 (b), we show the relationships between inference cost and performance on the examples of MATH500 Hendrycks et al. (2021). These examples have the corresponding difficulty-level annotations. We apply ABPO to DeepSeek-R1-Distill-Qwen-1.5B and evaluate its inference cost and performance on the subsets of different difficulty levels. From the figure, we observe that our ABPO can help the LLM generate CoT with proper lengths related to the example difficulties. This case shows that our ABPO can adaptively allocate more suitable budgets to more challenging examples during training, thereby achieving a good trade-off between performance and efficiency.

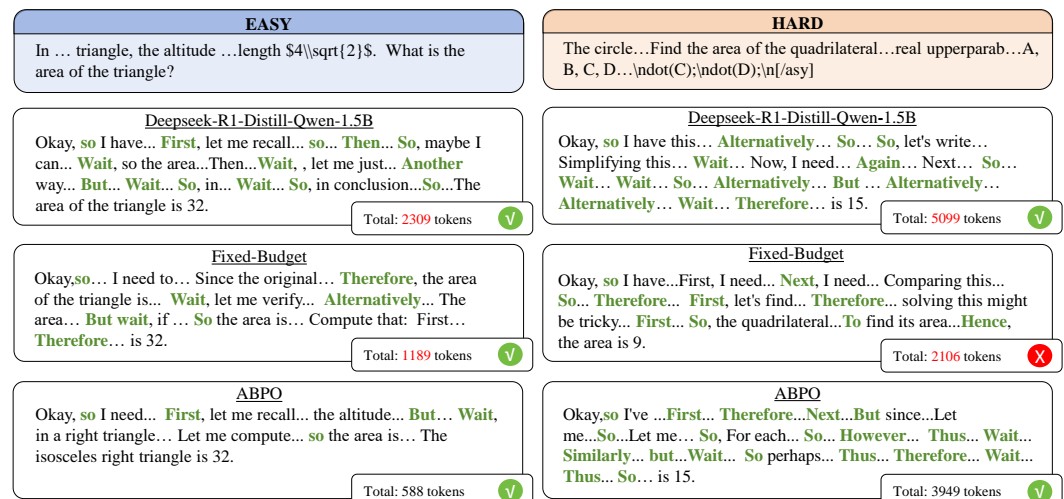

Figure 4: Visualization of CoTs and answers on both easy and hard examples by the default LLM, fixed-budget CoT optimization ($2k$ tokens) and our ABPO. These examples show that ABPO can help the LLM generate more concise and compact CoTs for simple questions and more complex and longer CoTs for the hard ones, thereby balancing the efficiency and performance of CoT reasoning.

## 4.4 QUALITATIVE ANALYSIS

To gain insights in to the proposed ABPO, we visualize its outputs on both easy and hard examples, and compare them with the default LLM and the fixed-budget baselines ($2k$), as shown in Fig. 4. The details of these examples can be found in Appendix A.4. From these examples, we can see that ABPO can generate concise and compact CoT traces on easy examples, helping the LLM quickly arrive at the correct answer without unnecessary elaboration. In contrast, the baseline model DeepSeek-R1-Distill-Qwen-1.5B tends to generate lengthy and redundant CoTs even for easy examples, leading to inefficient reasoning traces. For more difficult examples, ABPO adaptively generates longer CoTs that involve multiple attempts, self-reflection and alternative solution paths before giving the correct answer. Meanwhile, when training the LLM under fixed-budget settings, the reasoning process for the harder examples is too tightly to correctly answer the question. In other words, producing shorter CoTs will fail to contain the necessary reasoning steps, and also results in incorrect answers. This case demonstrates that ABPO is capable of adaptively performing test time scaling according to example difficulty, validating its ability to achieve difficulty-aware and efficient reasoning. Overall, these findings demonstrate the effectiveness of ABPO in adjusting token budgets according to example difficulty and its adaptive design principle.

## 5 CONCLUSION

In this paper, we focus on addressing the challenges of efficient and adaptive chain-of-thought (CoT) reasoning for *large language models* (LLMs), and propose a novel reinforcement learning scheme termed *Adaptive-Budgeting based Policy Optimization* (**ABPO**). To balance the efficiency and performance of CoT reasoning, ABPO introduces dynamic example pools to categorize training examples into three types based on the model's status, and then adopts an adaptive scheduling mechanism to assign proper token budgets for different examples during RL training. In this way, ABPO help the LLM to be capable of solving different tasks with adaptive CoT lengths, achieving a remarkable balance between the efficiency and performance. Extensive experiments on three representative LLMs and a bunch of CoT reasoning benchmarks are conducted, of which the results demonstrate that our ABPO method can achieve substantial efficiency improvements with minimal performance loss. Moreover, we also show obvious advantages over the compared methods in balancing the efficiency and performance of CoT reasoning. For instance, ABPO reduces token length by 78.3% while improving accuracy by 2.3% on DeepSeek-R1-Distill-Qwen-1.5B on average, and achieves 59.4% shorter reasoning length with 8.3% higher accuracy compared to HAPO. These results well confirm the effectiveness of ABPO towards efficient CoT reasoning for LLMs.

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

# A APPENDIX

## A.1 THE USE OF LARGE LANGUAGE MODELS (LLMS)

We employed a large language model (LLM) to assist with language polishing and proofreading. The use of the LLM was limited to improving clarity and readability of the manuscript, without influencing the design of experiments, analyses, or conclusions.

## A.2 PSEUDO-CODE OF ADAPTIVE-BUDGETING BASED POLICY OPTIMIZATION (ABPO)

In this subsection, we present the pseudo-code of ABPO.

---

**Algorithm 1:** Adaptive-Budgeting based Policy Optimization (ABPO)

---

**Input:** $D = \{(q_i, y_i)\}_{i=1}^N$
**Output:** LLM $M$ with compact CoT

1 **Init:** $\mathcal{S}_{\text{hard}} \leftarrow D$, $\mathcal{S}_{\text{learning}} \leftarrow \emptyset$, $\mathcal{S}_{\text{mastered}} \leftarrow \emptyset$; $\texttt{budget}[q] \leftarrow 0$; $t \leftarrow 0$; $k \leftarrow 1$.
2 **while** $t < S_{\max}$ **and** $\mathcal{S}_{mastered} \neq D$ **do**
3    **foreach** $q \in \mathcal{S}_{hard}$ **do**
4       $\texttt{budget}[q] \leftarrow t_0 + (k-1)d$;
5       **if** $Acc(q; n) > \tau_0$ **then** move $q$ to tail of $\mathcal{S}_{\text{learning}}$; ;
6    $L_0 \leftarrow |\mathcal{S}_{\text{learning}}|$;
7    **while** $|\mathcal{S}_{learning}| \geq \lambda L_0$ **and** $t < S_{\max}$ **do**
8       Sample batch $\mathcal{B} := \text{mix}(\mathcal{S}_{\text{mastered}}, \mathcal{S}_{\text{learning}}; \alpha)$ ;
9       GRPO_UPDATE$(M, \mathcal{B}, n)$; ;
10       **foreach** $q \in \mathcal{B}$ **do**
11          **if** $Acc(q; n) \geq \tau_1$ **then** move $q$ to tail of $\mathcal{S}_{\text{mastered}}$;
12          **else** move $q$ to tail of $\mathcal{S}_{\text{learning}}$;
13       $t \leftarrow t+1$;
14    $k \leftarrow k+1$;

---

## A.3 HYPERPARAMETER SETTINGS

Table 5 summarizes the detailed hyperparameter settings used in all ABPO experiments, including training steps, initial reasoning budgets $t_0$, thresholds $\tau_0$ and $\tau_1$, rollouts, and batch sizes for different models and data scales.

Table 5: Hyperparameter settings of ABPO experiments across different models and data scales.

| Model | Steps | $t_0$ | $d$ | $\tau_0$ | $\tau_1$ | Rollouts | Batch Size |
|---|---|---|---|---|---|---|---|
| DeepSeek-R1-Distill-Qwen-1.5B | 350 | 2000 | 500 | 0.0 | 0.85 | 16 | 128 |
| DeepScaleR-1.5B-Preview | 50 | 2000 | 500 | 0.2 | 0.85 | 16 | 128 |
| DeepSeek-R1-Distill-Qwen-7B | 50 | 3000 | 500 | 0.3 | 0.85 | 16 | 128 |
| DeepSeek-R1-Distill-Qwen-1.5B (32k data) | 350 | 2000 | 500 | 0.2 | 0.85 | 8 | 1024 |

## A.4 ADDITIONAL CASE STUDIES OF ABPO

We provide more detailed outputs on both easy and hard problems in Fig.4. The transition words are highlighted in red.

---

**EASY**

In a certain isosceles right triangle, the altitude to the hypotenuse has length $4\sqrt{2}$. What is the area of the triangle?

---

DeepSeek-R1-Distill-Qwen-1.5B

Okay, **so** I have this problem here: In a certain isosceles right triangle, the altitude to the hypotenuse has length $4\sqrt{2}$. I need to find the area of the triangle. Hmm, let's see. I remember that in an isosceles right triangle, the two legs are equal, and the angles are 45°, 45°, and 90°. **So**, maybe I can use some properties of right triangles here.

First, let me visualize the triangle. It's an isosceles right triangle, **so** the legs are equal. Let me denote the length of each leg as $x$. Then, the hypotenuse can be found using the Pythagorean theorem. Since it's a right-angled triangle, the hypotenuse $c$ is $x\sqrt{2}$. That makes sense because in a 45-45-90 triangle, the hypotenuse is leg length times $\sqrt{2}$.

Now, the problem mentions the altitude to the hypotenuse. In a right triangle, the altitude to the hypotenuse relates to the two segments it creates on the hypotenuse. I remember that the area can be calculated in two ways: one as half the product of the legs, and another using the hypotenuse and the altitude. **So**, maybe I can set up an equation using these two expressions for the area.

Let me denote the altitude to the hypotenuse as $h$, which is given as $4\sqrt{2}$. **So**, the area of the triangle is also equal to $\frac{1}{2} \times \text{hypotenuse} \times h$. Since I know the hypotenuse is $x\sqrt{2}$, substituting in the values, the area is $\frac{1}{2} \times x\sqrt{2} \times 4\sqrt{2}$.

Let me compute that: $\frac{1}{2} \times x\sqrt{2} \times 4\sqrt{2}$. Multiplying the constants first: $\frac{1}{2} \times 4 = 2$. Then, $\sqrt{2} \times \sqrt{2} = 2$. **So**, putting it together, the area becomes $2 \times x \times 2 = 4x$.

**Wait**, **so** the area is $4x$. **But** I also know that the area is $\frac{1}{2} \times \text{leg} \times \text{leg}$, which is $\frac{1}{2}x^2$. **So**, setting these two expressions equal: $4x = \frac{1}{2}x^2$. Hmm, let me solve for $x$.

Multiplying both sides by 2 to eliminate the fraction: $8x = x^2$. Then, bringing all terms to one side: $x^2 - 8x = 0$. Factoring out an $x$: $x(x - 8) = 0$. **So**, $x = 0$ or $x = 8$. Since a triangle can't have a side of length 0, $x = 8$.

Okay, **so** each leg is 8 units long. Then, to find the area, I can use $\frac{1}{2} \times x^2$ which is $\frac{1}{2} \times 8^2 = \frac{1}{2} \times 64 = 32$.

lternatively, using the other area expression, $4x$, **so** $4 \times 8 = 32$. Either way, I get the same result. **So**, the area is 32.

**Wait**, let me just make sure I didn't skip any steps or make a mistake. Starting from the beginning: it's an isosceles right triangle, legs equal, hypotenuse $x\sqrt{2}$. Altitude to hypotenuse is $4\sqrt{2}$. Area is $\frac{1}{2} \times \text{hypotenuse} \times \text{altitude}$, which is $\frac{1}{2} \times x\sqrt{2} \times 4\sqrt{2}$. Simplify: $\frac{1}{2} \times 4 = 2$, $\sqrt{2} \times \sqrt{2} = 2$, **so** 2 * x * 2 = 4x. Area is also $\frac{1}{2}x^2$. **So**, 4x = $\frac{1}{2}x^2$. Multiply both sides by 2: 8x = x^2. **So**, x^2 - 8x = 0. x(x - 8) = 0. **So**, x = 8. Then, area is $\frac{1}{2}x^2 = 32$. Yep, that seems right.

**Wait**, let me just make sure I didn't skip any steps or make a mistake. Starting from the beginning: it's an isosceles right triangle, legs equal, hypotenuse $x\sqrt{2}$. Altitude to hypotenuse is $4\sqrt{2}$. Area is $\frac{1}{2} \times$ hypotenuse $\times$ altitude, which is $\frac{1}{2} \times x\sqrt{2} \times 4\sqrt{2}$. Simplify: $\frac{1}{2} \times 4 = 2$, $\sqrt{2} \times \sqrt{2} = 2$, **so** 2 * x * 2 = 4x. Area is also $\frac{1}{2}x^2$. **So**, $4x = \frac{1}{2}x^2$. Multiply both sides by 2: $8x = x^2$. **So**, $x^2 - 8x = 0$. $x(x - 8) = 0$. **So**, x = 8. Then, area is $\frac{1}{2}x^2 = 32$. Yep, that seems right.

Another way to think about it: In an isosceles right triangle, the altitude to the hypotenuse also acts as a median and an angle bisector. Moreover, in such triangles, the altitude is equal to half the hypotenuse times $\sqrt{2}$. **Wait**, is that correct? Let me see.

If the legs are each $x$, then the hypotenuse is $x\sqrt{2}$. The altitude to the hypotenuse in a right triangle is given by $h = \frac{\text{product of legs}}{\text{hypotenuse}} = \frac{x \times x}{x\sqrt{2}} = \frac{x}{\sqrt{2}}$. **So**, $h = \frac{x}{\sqrt{2}}$.

**But** we are given $h = 4\sqrt{2}$, **so** setting them equal: $\frac{x}{\sqrt{2}} = 4\sqrt{2}$. Multiplying both sides by $\sqrt{2}$: $x = 4\sqrt{2} \times \sqrt{2} = 4 \times 2 = 8$. Yep, same result. **So**, legs are 8, hypotenuse is $8\sqrt{2}$, area is $32$. **So**, either way, the area is 32.

**Wait**, maybe I can also think about similar triangles here. In an isosceles right triangle, when you draw the altitude to the hypotenuse, it creates two smaller isosceles right triangles, each similar to the original triangle. **So**, each of these smaller triangles also has legs equal to each other. Let me try to confirm.

Let me denote the original triangle as ABC, right-angled at C, with legs AC and BC each of length $x$, and hypotenuse AB of length $x\sqrt{2}$. The altitude from C to AB is CD, which is given as $4\sqrt{2}$. Then, triangles ACD and BCD are both isosceles right triangles. **So**, in triangle ACD, legs AD and CD are equal. Similarly, in triangle BCD, legs BD and CD are equal.

**So**, in triangle ACD, legs AD and CD are equal, **so** AD = CD = $4\sqrt{2}$. Similarly, BD = CD = $4\sqrt{2}$. **Therefore**, the hypotenuse AB is AD + BD = $4\sqrt{2} + 4\sqrt{2} = 8\sqrt{2}$, which matches our earlier calculation that the hypotenuse is $x\sqrt{2}$, **so** $x = 8$. **So**, that's another verification.

**Alternatively**, using proportions: Since triangle ACD is similar to triangle ABC, the ratio of corresponding sides is the same. **So**, in triangle ACD, legs are AD = $4\sqrt{2}$, hypotenuse AC = 8 (since AC is a leg of the original triangle, **but wait**, no: AC is a leg of the original triangle, **but** CD is a leg in triangle ACD.

**Wait**, maybe this approach is getting a bit convoluted. Maybe better to stick with the two methods we already used, which both lead to the same answer.

**So**, in conclusion, the legs are each 8, hypotenuse is $8\sqrt{2}$, area is $\frac{1}{2} \times 8 \times 8 = 32$. **Alternatively**, using the altitude, $\frac{1}{2} \times 8\sqrt{2} \times 4\sqrt{2} = \frac{1}{2} \times 8 \times 4 \times 2 = 32$. **So**, both methods confirm the area is 32.

I think that's solid. I don't see any mistakes in my reasoning. **So**, I feel confident that the area is 32.

**Final Answer**
The area of the triangle is \boxed{32}.

756
757
758
759
760
761
762
763
764
765
766
767
768
769
770
771
772
773
774
775
776
777
778
779
780
781
782
783
784
785
786
787
788
789
790
791
792
793
794
795
796
797
798
799
800
801
802
803
804
805
806
807
808
809

---

### Fixed-Budget

Okay, **so** I have this problem about an isosceles right triangle where the altitude to the hypotenuse is $4\sqrt{2}$. I need to find the area of the triangle. Let me think about how to approach this.

First, let me recall what an isosceles right triangle looks like. It has two legs of equal length and a hypotenuse. The two non-right angles are each 45 degrees.

In such a triangle, the altitude to the hypotenuse is a line segment from the right angle to the hypotenuse, dividing it into two smaller congruent triangles. Since the original triangle is isosceles, I can use the properties of similar triangles here.

I remember that in an isosceles right triangle, the legs are equal, let's say each leg has length $a$. Then the hypotenuse will be $a\sqrt{2}$, because for a right triangle, the hypotenuse squared is equal to the sum of the squares of the other two sides.

Now, the altitude to the hypotenuse in a right triangle relates to the legs. I think the formula for the altitude $h$ is $h = \frac{a \cdot b}{c}$, where $a$ and $b$ are the legs, and $c$ is the hypotenuse. Since it's an isosceles triangle, $a = b$, **so** the formula simplifies to $h = \frac{a^2}{c}$.

Given $h = 4\sqrt{2}$, **so** substituting:

$4\sqrt{2} = \frac{a^2}{c}$

**But** $c = a\sqrt{2}$, **so** substitute that in:

$4\sqrt{2} = \frac{a^2}{a\sqrt{2}}$

Simplify the denominator:

$4\sqrt{2} = \frac{a^2}{a\sqrt{2}} = \frac{a}{\sqrt{2}}$

Hence:

$4\sqrt{2} = \frac{a}{\sqrt{2}}$

Multiply both sides by $\sqrt{2}$:

$4\sqrt{2} \cdot \sqrt{2} = a$

Simplify the left side:

$\sqrt{2} \cdot \sqrt{2} = 2$, **so**:

$4 \times 2 = a$

$a = 8$

**So** each leg is 8 units long. **Therefore**, the area of the triangle is $\frac{1}{2} \times a \times b$, which is $\frac{1}{2} \times 8 \times 8$.

---

810
811
812
813
814
815
816
817
818
819
820
821
822
823
824
825
826
827
828
829
830
831
832
833
834
835
836
837
838
839
840
841
842
843
844
845
846
847
848
849
850
851
852
853
854
855
856
857
858
859
860
861
862
863

Calculate that:

$\frac{1}{2} \times 64 = 32$

**So** the area is 32.

**Wait**, let me verify if my reasoning is correct.

**Alternatively**, I can think about the area in terms of the altitude. The area of the triangle can also be expressed as $\frac{1}{2} \times c \times h$, where $c$ is the hypotenuse and $h$ is the altitude.

We found that $c = a\sqrt{2}$, which is $8\sqrt{2}$.

**But wait**, if $a = 8$, then $c = 8\sqrt{2}$, which is correct.

**So** the area is $\frac{1}{2} \times 8\sqrt{2} \times 4\sqrt{2}$.

Compute that:

First, multiply the constants: $\frac{1}{2} \times 8 \times 4 = \frac{1}{2} \times 32 = 16$

Then multiply the radicals: $\sqrt{2} \times \sqrt{2} = 2$

**So** overall: $16 \times 2 = 32$. Perfect, same answer.

**Therefore**, the area of the triangle is 32.
</think>

The problem involves an isosceles right triangle with an altitude to the hypotenuse of length $4\sqrt{2}$. We need to find the area of the triangle.

1. **Identify the properties of the isosceles right triangle:**
   - Two legs of equal length, say $a$.
   - Hypotenuse $c = a\sqrt{2}$.

2. **Formula for the altitude to the hypotenuse:**
   - The altitude $h$ in a right triangle is given by $h = \frac{a \cdot b}{c}$.
   - Since $a = b$, this simplifies to $h = \frac{a^2}{c}$.

3. **Substitute the known values:**
   - Given $h = 4\sqrt{2}$ and $c = a\sqrt{2}$.
   - **So**, $4\sqrt{2} = \frac{a^2}{a\sqrt{2}}$.
   - Simplify: $4\sqrt{2} = \frac{a}{\sqrt{2}}$.
   - Multiply both sides by $\sqrt{2}$: $4 \times 2 = a$, **so** $a = 8$.

4. **Calculate the area:**
   - The area is $\frac{1}{2} \times a \times b = \frac{1}{2} \times 8 \times 8 = 32$.

Thus, the area of the triangle is $\boxed{32}$

864
865
866
867
868
869
870
871
872
873
874
875
876
877
878
879
880
881
882
883
884
885
886
887
888
889
890
891
892
893
894
895
896
897
898
899
900
901
902
903
904
905
906
907
908
909
910
911
912
913
914
915
916
917

ABPO

Okay, **so** I need to find the area of an isosceles right triangle where the altitude to the hypotenuse is 4√2. Hmm, let's see. First, let me recall some properties of isosceles right triangles.

An isosceles right triangle has two legs of equal length and a hypotenuse. The angles are 45°, 45°, and 90°. The legs are equal, let's call each leg 'a'. The hypotenuse in such a triangle is a√2.

Now, the altitude to the hypotenuse is given as 4√2. I remember that in a right triangle, the altitude to the hypotenuse relates to the lengths of the segments it creates on the hypotenuse. **But** in an isosceles right triangle, the altitude might have a special relationship.

**Wait**, in a right triangle, the length of the altitude to the hypotenuse can be calculated by the formula: (product of the legs) divided by the hypotenuse. **So**, in this case, the altitude h is (a * a) / (a√2). That simplifies to a² / (a√2) which is a / √2.

Given that this altitude is equal to 4√2, **so**:

a / √2 = 4√2

Let me solve for 'a' here. Multiply both sides by √2:

a = 4√2 * √2

√2 * √2 is 2, **so**:

a = 4 * 2 = 8

**So** each leg of the triangle is 8 units long.

Now, the area of the triangle is (base * height) / 2. Since it's an isosceles right triangle, both legs are equal, **so** the area is (a * a) / 2. Plugging in a = 8:

Area = (8 * 8) / 2 = 64 / 2 = 32

**So**, the area of the triangle is 32.

**Alternatively**, I can think in terms of the hypotenuse. The hypotenuse of the triangle is a√2 = 8√2. The altitude is given as 4√2. The area can also be expressed as (hypotenuse * altitude) / 2. **So**:

Area = (8√2 * 4√2) / 2

Multiply the terms:

(8 * 4) * (√2 * √2) / 2 = 32 * 2 / 2 = 32. Same result.

**So**, I'm confident the area is 32.
</think>

The area of the isosceles right triangle is \boxed{32}.

918
919
920
921
922
923
924
925
926
927
928
929
930
931
932
933
934
935
936
937
938
939
940
941
942
943
944
945
946
947
948
949
950
951
952
953
954
955
956
957
958
959
960
961
962
963
964
965
966
967
968
969
970
971

**HARD**

The circle $x^2 + y^2 = 2$ and the parabola $y^2 = 8x$ have two common tangents, forming four points of tangency. Find the area of the quadrilateral formed by the four points of tangency.

[asy] unitsize(0.8 cm);
real upperparab (real x) {return (sqrt(8*x));} real lowerparab (real x) {return (-sqrt(8*x));}
pair A, B, C, D; A = (-1,1); B = (2,4); C = (-1,-1); D = (2,-4);
draw(graph(upperparab,0,3)); draw(graph(lowerparab,0,3)); draw(Circle((0,0),sqrt(2)));
draw(interp(A,B,-0.2)--interp(A,B,1.2)); draw(interp(C,D,-0.2)--interp(C,D,1.2));
draw(A--C); draw(B--D); dot(A); dot(B); dot(C); dot(D); [/asy]

---

DeepSeek-R1-Distill-Qwen-1.5B

Okay, **so** I have this problem where I need to find the area of a quadrilateral formed by the points where two common tangents to a circle and a parabola touch each curve. The circle is given by the equation $x^2 + y^2 = 2$ and the parabola is $y^2 = 8x$. The figure provided in the Asymptote code shows the circle and the parabola, and then the tangents which intersect at four points forming a quadrilateral.

First, I need to understand the problem. We have a circle and a parabola, and we're supposed to find their common tangents. Each tangent will touch both the circle and the parabola at exactly one point. Since there are two common tangents, each will have two points of tangency, and these four points will form a quadrilateral. The task is to find the area of this quadrilateral.

I know that a common tangent to two curves is a line that touches each curve at exactly one point. For a circle and a parabola, the equations of the common tangents can be found by solving certain conditions. The approach is likely to involve setting up equations for the tangent lines and then solving for the points where these lines are tangent to both the circle and the parabola.

Let me start by recalling that the equation of a tangent to a circle $x^2 + y^2 = r^2$ at a point $(x_1, y_1)$ is $xx_1 + yy_1 = r^2$. In this case, the circle has radius $\sqrt{2}$, **so** the tangent equation would be $xx_1 + yy_1 = 2$.

For the parabola $y^2 = 8x$, which opens to the right, the equation of a tangent can be written in the form $y = mx + \frac{2}{m}$, where $m$ is the slope of the tangent. **Alternatively**, the tangent can be expressed as $y = mx + c$, and for the parabola $y^2 = 4ax$, the condition for the line $y = mx + c$ to be tangent is $c = \frac{a}{m}$. In this case, since $4a = 8$, $a = 2$, **so** $c = \frac{2}{m}$. That seems right.

**So**, for the parabola $y^2 = 8x$, the tangent equations will be $y = mx + \frac{2}{m}$.

Now, since these lines are also tangent to the circle $x^2 + y^2 = 2$, the distance from the center of the circle (which is at (0,0)) to the tangent line must be equal to the radius of the circle, which is $\sqrt{2}$.

The formula for the distance from a point $(x_0, y_0)$ to the line $ax + by + c = 0$ is $\frac{|ax_0 + by_0 + c|}{\sqrt{a^2 + b^2}}$. **So**, let's write the equation of the tangent line in standard form.

Given $y = mx + \frac{2}{m}$, we can rearrange it to $mx - y + \frac{2}{m} = 0$. **So,** $a = m$, $b = -1$, and $c = \frac{2}{m}$.

The distance from (0,0) to this line is $\frac{|0 + 0 + \frac{2}{m}|}{\sqrt{m^2 + (-1)^2}}} = \frac{|\frac{2}{m}|}{\sqrt{m^2 + 1}}}$.

Since this distance must equal the radius $\sqrt{2}$, we have:

$\frac{\frac{2}{|m|}}{\sqrt{m^2 + 1}} = \sqrt{2}$

Simplifying this, we can write:

$\frac{2}{|m| \sqrt{m^2 + 1}} = \sqrt{2}$

Multiply both sides by $|m| \sqrt{m^2 + 1}$:

$2 = \sqrt{2} |m| \sqrt{m^2 + 1}$

Divide both sides by $\sqrt{2}$:

$\sqrt{2} = |m| \sqrt{m^2 + 1}$

Square both sides to eliminate the square roots:

$2 = m^2 (m^2 + 1)$

Expanding the right side:

$2 = m^4 + m^2$

Bring all terms to one side:

$m^4 + m^2 - 2 = 0$

This is a quadratic in terms of $m^2$. Let me set $u = m^2$, **so** the equation becomes:

$u^2 + u - 2 = 0$

Solving this quadratic equation using the quadratic formula:

$u = \frac{-1 \pm \sqrt{1 + 8}}{2} = \frac{-1 \pm 3}{2}$

**So**, $u = \frac{2}{2} = 1$ or $u = \frac{-4}{2} = -2$. Since $u = m^2$ must be non-negative, we discard $u = -2$. Thus, $u = 1$, which means $m^2 = 1$, **so** $m = \pm 1$.

**So**, the slopes of the common tangents are $m = 1$ and $m = -1$.

**Therefore**, the equations of the tangent lines are:

For $m = 1$: $y = x + \frac{2}{1} = x + 2$

For $( m = -1 )$: $( y = -x + \frac{2}{-1} = -x - 2 )$

**Wait**, hold on, earlier I wrote the tangent equations as $( y = mx + \frac{2}{m} )$. **So**, for $( m = 1 )$, it's $( y = x + 2 )$, and for $( m = -1 )$, it's $( y = -x - 2 )$.

**But** let's verify this because sometimes the signs can be tricky.

**Wait**, when $( m = -1 )$, the tangent equation would be $( y = -x + \frac{2}{-1} = -x - 2 )$. That seems correct.

Now, I need to find the points where these tangent lines touch the circle and the parabola.

First, let's find the points of tangency on the parabola $( y^2 = 8x )$.

For the line $( y = x + 2 )$:

Substitute into the parabola equation:

$( (x + 2)^2 = 8x )$

Expanding:

$( x^2 + 4x + 4 = 8x )$

Bring all terms to one side:

$( x^2 - 4x + 4 = 0 )$

This factors as:

$( (x - 2)^2 = 0 )$

**So**, $( x = 2 )$. Then, $( y = 2 + 2 = 4 )$. **So**, the point of tangency on the parabola is (2, 4).

Similarly, for the line $( y = -x - 2 )$:

Substitute into the parabola equation:

$( (-x - 2)^2 = 8x )$

Expanding:

$( x^2 + 4x + 4 = 8x )$

Bring all terms to one side:

$( x^2 - 4x + 4 = 0 )$

**Again**, this factors as:

$( (x - 2)^2 = 0 )$

Thus, $( x = 2 )$, and $( y = -2 - 2 = -4 )$. **So**, the point of tangency on the parabola is (2, -4).

Now, let's find the points of tangency on the circle $( x^2 + y^2 = 2 )$.

For the line $( y = x + 2 )$:

Substitute into the circle equation:

$( x^2 + (x + 2)^2 = 2 )$

Expanding:

$( x^2 + x^2 + 4x + 4 = 2 )$

Combine like terms:

$( 2x^2 + 4x + 2 = 0 )$

Divide by 2:

$( x^2 + 2x + 1 = 0 )$

This factors as:

$( (x + 1)^2 = 0 )$

**So**, $( x = -1 )$. Then, $( y = -1 + 2 = 1 )$. **Therefore**, the point of tangency on the circle is (-1, 1).

Similarly, for the line $( y = -x - 2 )$:

Substitute into the circle equation:

$( x^2 + (-x - 2)^2 = 2 )$

Expanding:

$( x^2 + x^2 + 4x + 4 = 2 )$

Combine like terms:

$( 2x^2 + 4x + 2 = 0 )$

Divide by 2:

$( x^2 + 2x + 1 = 0 )$

Factor as:

$( (x + 1)^2 = 0 )$

**So**, $( x = -1 )$. Then, $( y = 1 - 2 = -1 )$. **Therefore**, the point of tangency on the circle is (-1, -1).

**So**, the four points of tangency are:

1. (-1, 1) on the circle
2. (2, 4) on the parabola
3. (-1, -1) on the circle
4. (2, -4) on the parabola

Now, I need to find the area of the quadrilateral formed by connecting these four points: (-1, 1), (2, 4), (-1, -1), and (2, -4).

Let me plot these points mentally. (-1, 1) is in the second quadrant, (2, 4) is in the first quadrant, (-1, -1) is in the third quadrant, and (2, -4) is in the fourth quadrant.

Connecting these points in order would form a quadrilateral. To find its area, I can use the shoelace formula, which is a method for finding the area of a polygon when the coordinates of the vertices are known.

The shoelace formula states that for a polygon with vertices (x1, y1), (x2, y2), ..., (xn, yn), the area is:

$( \frac{1}{2} | \sum_{i=1}^{n} (x_i y_{i+1} - x_{i+1} y_i) | )$

where (xn+1, yn+1) is (x1, y1).

**So**, let's list the coordinates in order. Let's go around the quadrilateral clockwise or counterclockwise. Let me choose the order: (-1, 1), (2, 4), (2, -4), (-1, -1), and back to (-1, 1).

Let me write them down:

1. (-1, 1)
2. (2, 4)
3. (2, -4)
4. (-1, -1)
5. (-1, 1)  // Closing the polygon

Now, applying the shoelace formula:

First, compute the sum of $( x_i y_{i+1} )$:

- (-1) * 4 = -4
- 2 * (-4) = -8
- 2 * (-1) = -2
-   (-1) * 1 = -1

Sum: (-4) + (-8) + (-2) + (-1) = -15

Next, compute the sum of $( y_i x_{i+1} )$:

- 1 * 2 = 2
- 4 * 2 = 8
- (-4) * (-1) = 4
- (-1) * (-1) = 1

Sum: 2 + 8 + 4 + 1 = 15

Now, subtract the two sums:

-15 - 15 = -30

Take the absolute value and divide by 2:

\( \frac{1}{2} | -30 | = \frac{1}{2} * 30 = 15 \)

**So**, the area is 15.

**Wait**, that seems straightforward, **but** let me double-check my calculations because sometimes the order of the points can affect the result.

**Alternatively**, I can plot the points and see if the quadrilateral is a trapezoid or some other shape.

Looking at the coordinates:

(-1, 1), (2, 4), (2, -4), (-1, -1)

Plotting these, we see that two points are at (-1, ±1) and (2, ±4). **So**, the quadrilateral is symmetric with respect to both the x-axis and y-axis. **Wait**, is it symmetric?

**Wait**, (-1, 1) and (-1, -1) are symmetric over the x-axis. Similarly, (2,4) and (2, -4) are symmetric over the x-axis. Similarly, the points (-1,1) and (2,4) are not symmetric, as are (-1,-1) and (2,-4). **So**, **perhaps** the quadrilateral is symmetric across both axes.

**But** regardless, the shoelace formula should work as long as the points are ordered correctly, either clockwise or counterclockwise.

Let me try a different order to double-check.

Let me arrange the points in a different order, say: (-1, 1), (-1, -1), (2, -4), (2, 4), and back to (-1, 1).

Computing shoelace formula **again**:

First sum:

- (-1) * (-1) = 1
- (-1) * (-4) = 4
- 2 * 4 = 8
- 2 * 1 = 2

Sum: 1 + 4 + 8 + 2 = 15

Second sum:

- 1 * (-1) = -1
- (-1) * 2 = -2
- (-4) * 2 = -8
- 4 * (-1) = -4

Sum: (-1) + (-2) + (-8) + (-4) = -15

Subtract: 15 - (-15) = 30

Take absolute value and divide by 2: 15

Same result, **so** that's consistent.

**Wait**, actually, the shoelace formula requires the points to be ordered either clockwise or counterclockwise without crossing. **So**, the first ordering I did was (-1,1), (2,4), (2,-4), (-1,-1), which goes around the quadrilateral in clockwise order, and the second was (-1,1), (-1,-1), (2,-4), (2,4), which is also clockwise.

Either way, the area is 15.

**Alternatively**, maybe we can think of the quadrilateral as a trapezoid. Let me see.

**Wait**, looking at the coordinates, the quadrilateral is symmetric across the x-axis. **So**, if I consider the upper half and the lower half separately.

**But perhaps** that complicates things because it's a trapezoid.

**Alternatively**, think of the quadrilateral as a polygon with vertices at (-1,1), (2,4), (2,-4), (-1,-1). If I connect these points, I can divide it into two triangles.

**Alternatively**, think of it as a kite-shaped quadrilateral because two pairs of adjacent sides are equal. **Wait**, is that the case?

Compute the distances between consecutive points:

1. From (-1,1) to (2,4):

Distance squared: (2 - (-1))^2 + (4 - 1)^2 = (3)^2 + (3)^2 = 9 + 9 = 18

2. From (2,4) to (2,-4):

Distance squared: (2 - 2)^2 + (-4 - 4)^2 = 0 + (-8)^2 = 64

3. From (2,-4) to (-1,-1):

Distance squared: (-1 - 2)^2 + (-1 - (-4))^2 = (-3)^2 + (3)^2 = 9 + 9 = 18

4. From (-1,-1) to (-1,1):

Distance squared: (-1 - (-1))^2 + (1 - (-1))^2 = 0 + (2)^2 = 4

**So**, sides are sqrt(18), 8, sqrt(18), and 2.

**So**, the sides are not equal, **so** it's not a kite. Maybe it's a trapezoid? Let me check if any sides are parallel.

Compute the slopes of the sides:

1. From (-1,1) to (2,4): slope is (4 - 1)/(2 - (-1)) = 3/3 = 1.

2. From (2,4) to (2,-4): slope is undefined (vertical line).

3. From (2,-4) to (-1,-1): slope is (-1 - (-4))/(-1 - 2) = (3)/(-3) = -1.

4. From (-1,-1) to (-1,1): slope is undefined (vertical line).

**So**, sides 2 and 4 are both vertical, hence parallel. Sides 1 and 3 have slopes 1 and -1, which are not equal, **so** they are not parallel. **So**, it's a trapezoid with two sides parallel.

**Therefore**, the area can be found by the formula for the area of a trapezoid: $\frac{1}{2} (b_1 + b_2) h$, where $b_1$ and $b_2$ are the lengths of the two parallel sides, and $h$ is the height (distance between them).

Looking at the trapezoid, the two parallel sides are the vertical sides from (2,4) to (2,-4) and from (-1,1) to (-1,-1). Their lengths are:

- From (2,4) to (2,-4): distance is |4 - (-4)| = 8 units.
- From (-1,1) to (-1,-1): distance is |1 - (-1)| = 2 units.

**So**, $b_1 = 8$, $b_2 = 2$.

Now, the height $h$ is the horizontal distance between the two parallel sides. Since the sides are vertical at x=2 and x=-1, the distance between them is |2 - (-1)| = 3 units.

**Therefore**, area is $\frac{1}{2} (8 + 2) * 3 = \frac{1}{2} * 10 * 3 = 5 * 3 = 15$.

Same result as before.

**So**, regardless of the method, the area is 15. **Therefore**, I can be confident that 15 is the correct area of the quadrilateral.

**But wait**, before I conclude, let me just visualize the points **again**. The quadrilateral has two vertical sides at x=2 and x=-1, connecting (2,4) to (2, -4) and (-1,1) to (-1,-1). Then, the other two sides are the lines connecting (2,4) to (-1,-1) and (2,-4) to (-1,1). These cross each other, forming a quadrilateral.

**Alternatively**, I can compute the area using vectors or coordinate geometry. **But** since both methods gave me 15, and my reasoning is consistent, I think it's safe.

**Therefore**, the area of the quadrilateral formed by the four points of tangency is 15.

**Final Answer**
\boxed{15}
</think>

To find the area of the quadrilateral formed by the points of tangency of the common tangents to the circle $x^2 + y^2 = 2$ and the parabola $y^2 = 8x$, we start by determining the equations of the common tangents.

1. The circle $x^2 + y^2 = 2$ has radius $\sqrt{2}$.
2. The parabola $y^2 = 8x$ has the tangent equation $y = mx + \frac{2}{m}$.

Using the condition that the distance from the center of the circle (0,0) to the tangent line must equal the radius $\sqrt{2}$, we find the slopes $m$:

$$
\frac{\left| \frac{2}{m} \right|}{\sqrt{m^2 + 1}} = \sqrt{2}
$$

Solving this, we get $m = \pm 1$. Thus, the equations of the tangent lines are $y = x + 2$ and $y = -x - 2$.

Next, we find the points of tangency on the parabola and the circle:

- For the line $y = x + 2$:
  - Tangency on the parabola: $(2, 4)$
  - Tangency on the circle: $(-1, 1)$

- For the line $y = -x - 2$:
  - Tangency on the parabola: $(2, -4)$
  - Tangency on the circle: $(-1, -1)$

The four points of tangency are: (-1, 1), (2, 4), (-1, -1), and (2, -4).

Using the shoelace formula to find the area of the quadrilateral formed by these points:

1. List the coordinates in order: (-1, 1), (2, 4), (2, -4), (-1, -1).
2. Apply the shoelace formula:

$$
\text{Area} = \frac{1}{2} \left| (-1 \cdot 4 + 2 \cdot (-4) + 2 \cdot (-1) + (-1) \cdot 1) - (1 \cdot 2 + 4 \cdot 2 + (-4) \cdot (-1) + (-1) \cdot (-1)) \right|
$$

Calculating the sums:

- First sum: $-4 - 8 - 2 - 1 = -15$
- Second sum: $2 + 8 + 4 + 1 = 15$

Subtracting and taking the absolute value:

\[
\left| -15 - 15 \right| = 30
\]

Dividing by 2:

\[
\frac{1}{2} \times 30 = 15
\]

Thus, the area of the quadrilateral is:

\[
\boxed{15}
\]

1458
1459
1460
1461
1462
1463

---

Fixed-Budget

Okay, **so** I have this problem where I need to find the area of a quadrilateral formed by the points of tangency between the circle $( x^2 + y^2 = 2 )$ and the parabola $( y^2 = 8x )$. The figure provided is an Asymptote drawing, which shows the circle and the parabola, and some tangents with points of contact labeled A, B, C, D.

First, I need to recall how to find common tangents between a circle and a parabola. Since both are symmetric with respect to the x-axis, the points of tangency will also be symmetric. **So**, if point A is at (a, b), point C will be at (a, -b), and similarly for points B and D.

Next, I need an equation for the tangent lines to both the circle and the parabola. For a parabola $( y^2 = 4ax )$, the equation of the tangent at point $( (at^2, 2at) )$ is $( ty = x + at^2 )$. Comparing this with our given parabola $( y^2 = 8x )$, which can be seen as $( y^2 = 4a x )$ with $( a = 2 )$, **so** $( 4a = 8 )$.

**So**, the tangent equation becomes $( ty = x + 2t^2 )$.

Now, this line should also be tangent to the circle $( x^2 + y^2 = 2 )$. The condition for a line $( y = mx + c )$ to be tangent to a circle $( x^2 + y^2 = r^2 )$ is that $( c^2 = r^2(1 + m^2) )$.

Let me rearrange the tangent line equation:

$( ty = x + 2t^2 )$ --> $( x - ty + 2t^2 = 0 )$.

**So**, it's in the form $( Ax + By + C = 0 )$, where $( A = 1 )$, $( B = -t )$, $( C = 2t^2 )$.

**Therefore**, the condition for tangency is:

$( C^2 = r^2(1 + m^2) )$.

**But** I need to express this in terms of t.

First, let's find the slope m of the tangent line.

From the equation $( x - ty + 2t^2 = 0 )$, we can write it as $( y = \frac{1}{t}x + \frac{2t}{1} )$.

**So**, $( m = \frac{1}{t} )$.

**Therefore**, $( m^2 = \frac{1}{t^2} )$.

Now, plugging back into the tangency condition:

$( (2t^2)^2 = 2^2 \left(1 + \frac{1}{t^2}\right) )$.

Simplify:

$( 4t^4 = 4 \left(1 + \frac{1}{t^2}\right) )$.

Divide both sides by 4:

---

1508
1509
1510
1511

$( t^4 = 1 + \frac{1}{t^2} )$.

Multiply both sides by $( t^2 )$:

$( t^6 = t^2 + 1 )$.

This is a cubic equation in terms of $( z = t^2 )$:

$( z^3 - z - 1 = 0 )$.

Hmm, solving this might be tricky. Maybe I can find rational roots. Trying z=1: $( 1 -1 -1 = -1 \neq 0 )$. z= -1: negative, **so** no. **So**, probably no rational roots. Maybe I can use some approximation method or use the cubic formula, **but** that might be complicated.

**Alternatively**, maybe I can use substitution or numerical methods. **Alternatively**, **perhaps** there's a substitution.

Let me consider $( z = t^2 )$, **so** the equation is $( z^3 - z -1 =0 )$.

**Perhaps** I can use the method for solving cubics.

The general formula for a cubic $( z^3 + az^2 + bz + c = 0 )$ is:

$( z = \sqrt[3]{-\frac{c}{2} + \sqrt{\left(\frac{c}{2}\right)^2 + \left(\frac{a}{3}\right)^3}} + \sqrt[3]{-\frac{c}{2} - \sqrt{\left(\frac{c}{2}\right)^2 + \left(\frac{a}{3}\right)^3}} )$.

In our case, $( a = 0 )$, $( b = -1 )$, $( c = -1 )$.

**So**,

$( z = \sqrt[3]{\frac{1}{2} + \sqrt{\left(\frac{1}{2}\right)^2 + 0}} + \sqrt[3]{\frac{1}{2} - \sqrt{\left(\frac{1}{2}\right)^2 + 0}} )$.

Simplify:

$( z = \sqrt[3]{\frac{1}{2} + \frac{1}{2}} + \sqrt[3]{\frac{1}{2} - \frac{1}{2}} )$.

$( z = \sqrt[3]{1} + \sqrt[3]{0} = 1 + 0 =1 )$.

**So**, one real root is $( z =1 )$, **so** $( t^2=1 )$, **so** $( t=1 )$ or $( t=-1 )$.

**Therefore**, we can take t=1 or t=-1.

**So**, the tangent lines are:

For t=1:

$( y = \frac{1}{1}x + \frac{2(1)^2}{1} = x + 2 )$.

For t=-1:

$( y = \frac{1}{-1}x + \frac{2(-1)^2}{1} = -x + 2 )$.

**So**, the two tangent lines are $( y = x + 2 )$ and $( y = -x + 2 )$.

Now, let's find the points of contact on both the circle and the parabola.

For the parabola $( y^2 =8x )$, the points of tangency are $( (2t^2, 4t) )$.

**So** for t=1, point is $( (2(1)^2, 4(1)) = (2, 4) )$.

For t=-1, point is $( (2(-1)^2, 4(-1)) = (2, -4) )$.

**So**, the two points of tangency on the parabola are A(2,4) and B(2,-4).

Similarly, for the circle $( x^2 + y^2 = 2 )$, the points of tangency can be found by solving the system of the circle and the tangent line.

For line $( y = x + 2 )$:

$( x^2 + (x + 2)^2 = 2 )$.

$( x^2 + x^2 +4x +4 = 2 )$.

$( 2x^2 +4x +2 =0 )$.

Divide by 2:

$( x^2 +2x +1 =0 )$.

$( (x+1)^2 =0 )$.

**So**, $( x = -1 )$.

Then, $( y = -1 +2 =1 )$.

Thus, the point of tangency is (-1,1).

Similarly, for line $( y = -x +2 )$:

$( x^2 + (-x +2)^2 = 2 )$.

$( x^2 + x^2 -4x +4 =2 )$.

$( 2x^2 -4x +2 =0 )$.

Divide by 2:

$( x^2 -2x +1 =0 )$.

$( (x-1)^2 =0 )$.

Thus, $x = 1$, $y = -1 + 2 = 1$.

Thus, the point of tangency is (1,1).

**So**, the four points of tangency are:

A(2,4), B(2,-4), C(-1,1), D(1,1).

Now, we need to find the area of quadrilateral ABCD.

Given the points A(2,4), B(2,-4), C(-1,1), D(1,1).

First, I can plot these points:

A(2,4) is in the upper right quadrant.

B(2,-4) is in the lower right quadrant.

C(-1,1) is in the upper left quadrant.

D(1,1) is in the upper right quadrant.

**So**, the quadrilateral is symmetric with respect to the y-axis.

To find its area, I can use the shoelace formula.

The coordinates in order are A(2,4), B(2,-4), C(-1,1), D(1,1), back to A(2,4).

Compute shoelace sum:

Sum1 = (2)(-4) + (2)(1) + (-1)(1) + (1)(4) = (-8) + 2 + (-1) + 4 = (-8+2) + (-1+4) = (-6) + 3 = -3.

Sum2 = (4)(2) + (-4)(-1) + (1)(1) + (1)(2) = 8 + 4 + 1 + 2 = 15.

Area = (|Sum1 - Sum2|)/2 = | -3 -15 | /2 = | -18 | /2 = 9.

Hence, the area is 9.

1674
1675
1676
1677
1678
1679
1680
1681
1682
1683
1684
1685
1686
1687
1688
1689
1690
1691
1692
1693
1694
1695
1696
1697
1698
1699
1700
1701
1702
1703
1704
1705
1706
1707
1708
1709
1710
1711
1712
1713
1714
1715
1716
1717
1718
1719
1720
1721
1722
1723
1724
1725
1726
1727

ABPO

Okay, **so** I've got this problem where I need to find the area of a quadrilateral formed by the points of tangency of two common tangents to a circle and a parabola. The circle is given by the equation $x^2 + y^2 = 2$, and the parabola is $y^2 = 8x$. The points of tangency are four in total, and they form a quadrilateral. I need to find the area of this quadrilateral.

First, I remember that common tangents to a circle and a parabola are lines that touch both the circle and the parabola at exactly one point each. Since there are two common tangents, each will touch both the circle and the parabola, resulting in four points of tangency.

I think the Asymptote code provided might help me visualize the figure. It shows the circle centered at the origin with radius $\sqrt{2}$, and the parabola $y^2 = 8x$ opens to the right. The points A, B, C, D are given as (-1,1), (2,4), (-1,-1), and (2,-4). **So**, the quadrilateral is formed by connecting these points.

**But** I need to confirm this. Maybe I should derive the equations of the common tangents and then find their points of contact with the circle and parabola.

Let me start by recalling some properties. For a parabola $y^2 = 4ax$, the general equation of a tangent line is $y = tx + a/t$, where t is a parameter. Comparing this with $y^2 = 8x$, which is $y^2 = 4a x$ with $4a = 8$, **so** $a = 2$. Thus, the tangent equation becomes $y = tx + 2/t$.

Similarly, the circle $x^2 + y^2 = 2$ has a general tangent equation. The tangent to a circle $x^2 + y^2 = r^2$ at point $(x_1, y_1)$ is $xx_1 + yy_1 = r^2$. **So**, if I have a tangent line $y = mx + c$, then the condition for it to be tangent to the circle is $c^2 = r^2(1 + m^2)$. Here, $r^2 = 2$, **so** $c^2 = 2(1 + m^2)$.

**So**, the tangent lines to the circle will have the form $y = mx \pm \sqrt{2(1 + m^2)}$. These lines should also be tangent to the parabola $y^2 = 8x$.

Substituting $y = mx \pm \sqrt{2(1 + m^2)}$ into $y^2 = 8x$:

$(mx \pm \sqrt{2(1 + m^2)})^2 = 8x$

Expanding the left side:

$m^2x^2 \pm 2mx\sqrt{2(1 + m^2)} + 2(1 + m^2) = 8x$

This is a quadratic in x. For the line to be tangent to the parabola, the discriminant of this quadratic should be zero.

**So**, let me write it as:

$m^2x^2 \pm 2mx\sqrt{2(1 + m^2)} + 2(1 + m^2) - 8x = 0$

Arrange terms:

$m^2x^2 + (\pm 2m\sqrt{2(1 + m^2)} - 8)x + 2(1 + m^2) = 0$

For this quadratic equation, the discriminant D is:

D = [±2m√(2(1 + m²)) - 8]^2 - 4 * m² * 2(1 + m²) = 0

This seems a bit complicated, **but** let's compute it step by step.

First, compute the discriminant:

D = [±2m√(2(1 + m²)) - 8]^2 - 8m²(1 + m²) = 0

Let me compute the first square term:

[±2m√(2(1 + m²)) - 8]^2 = (±2m√(2(1 + m²)))^2 - 2 * 2m√(2(1 + m²)) * 8 + 8²

**Wait**, actually, expanding (a - b)^2 = a² - 2ab + b².

**So**,

= (2m√(2(1 + m²)))^2 - 2 * 2m√(2(1 + m²)) * 8 + 8²

= 4m² * 2(1 + m²) - 32m√(2(1 + m²)) + 64

= 8m²(1 + m²) - 32m√(2(1 + m²)) + 64

Then subtract 8m²(1 + m²):

**So**,

D = [8m²(1 + m²) - 32m√(2(1 + m²)) + 64] - 8m²(1 + m²) = 0

Simplify:

8m²(1 + m²) cancels with -8m²(1 + m²), leaving:

-32m√(2(1 + m²)) + 64 = 0

**So**,

-32m√(2(1 + m²)) + 64 = 0

Move 64 to the other side:

-32m√(2(1 + m²)) = -64

Divide both sides by -32:

m√(2(1 + m²)) = 2

Square both sides to eliminate the square root:

m² * 2(1 + m²) = 4

Simplify:

$2m^2(1 + m^2) = 4$

Divide both sides by 2:

$m^2(1 + m^2) = 2$

This is a quartic equation:

$m^4 + m^2 - 2 = 0$

Let me set $z = m^2$, **so** equation becomes:

$z^2 + z - 2 = 0$

Solve for z:

$z = [-1 \pm \sqrt{(1 + 8)}] / 2 = [-1 \pm 3]/2$

**So**, $z = (2)/2 = 1$ or $z = (-4)/2 = -2$

Since $z = m^2$ cannot be negative, $z = 1$.

Thus, $m^2 = 1 \Rightarrow m = \pm 1$

**Therefore**, the slopes of the common tangents are $m = 1$ and $m = -1$.

**So**, the equations of the common tangents are:

For $m = 1$: $y = x \pm \sqrt{(2(1 + 1))} = x \pm \sqrt{4} = x \pm 2$

For $m = -1$: $y = -x \pm \sqrt{(2(1 + 1))} = -x \pm 2$

**So**, the four lines are $y = x + 2$, $y = x - 2$, $y = -x + 2$, and $y = -x - 2$.

Now, I need to find the points of tangency on both the circle and the parabola.

Let's start with the circle $x^2 + y^2 = 2$.

For each tangent line, solve for the point of contact.

Take $y = x + 2$.

Substitute into circle:

$x^2 + (x + 2)^2 = 2$

$x^2 + x^2 + 4x + 4 = 2$

$2x^2 + 4x + 2 = 0$

Divide by 2:

$x^2 + 2x + 1 = 0$

This factors as $(x + 1)^2 = 0 => x = -1$

Then $y = -1 + 2 = 1$

**So**, one point is (-1, 1).

Similarly, for $y = x - 2$:

Substitute into circle:

$x^2 + (x - 2)^2 = 2$

$x^2 + x^2 - 4x + 4 = 2$

$2x^2 - 4x + 2 = 0$

Divide by 2:

$x^2 - 2x + 1 = 0 => (x - 1)^2 = 0 => x = 1$

Then $y = 1 - 2 = -1$

**So**, the point is (1, -1).

Similarly, for $y = -x + 2$:

Substitute into circle:

$x^2 + (-x + 2)^2 = 2$

$x^2 + x^2 - 4x + 4 = 2$

$2x^2 - 4x + 2 = 0$

Divide by 2:

$x^2 - 2x + 1 = 0 => (x - 1)^2 = 0 => x = 1$

Then $y = -1 + 2 = 1$

Point is (1, 1).

For $y = -x - 2$:

Substitute into circle:

$x^2 + (-x - 2)^2 = 2$

$x^2 + x^2 + 4x + 4 = 2$

$2x^2 + 4x + 2 = 0$

Divide by 2:

$x^2 + 2x + 1 = 0 \Rightarrow (x + 1)^2 = 0 \Rightarrow x = -1$

Then $y = 1 - 2 = -1$

Point is (-1, -1).

**So**, the four points of tangency on the circle are (-1,1), (1,-1), (1,1), (-1,-1).

Now, let's find the points of tangency on the parabola $y^2 = 8x$.

For each tangent line, solve for the point of contact.

Take $y = x + 2$.

Substitute into parabola:

$(x + 2)^2 = 8x$

$x^2 + 4x + 4 = 8x$

$x^2 - 4x + 4 = 0$

This factors as $(x - 2)^2 = 0 \Rightarrow x = 2$

Then $y = 2 + 2 = 4$. **So**, point is (2,4).

Similarly, $y = x - 2$:

$(x - 2)^2 = 8x$

$x^2 - 4x + 4 = 8x$

$x^2 - 12x + 4 = 0$

Solve using quadratic formula:

$x = [12 \pm \sqrt{(144 - 16)}]/2 = [12 \pm \sqrt{128}]/2 = [12 \pm 8\sqrt{2}]/2 = 6 \pm 4\sqrt{2}$

**Wait**, **but** then $y = x - 2$, **so** $y$ would be $(6 \pm 4\sqrt{2}) - 2 = 4 \pm 4\sqrt{2}$.

**But** since the parabola is $y^2 = 8x$, which opens to the right, the points $(6 + 4\sqrt{2}, 4 + 4\sqrt{2})$ and $(6 - 4\sqrt{2}, 4 - 4\sqrt{2})$ are valid.

**Wait**, **but** maybe I made a mistake because the tangent lines are $y = x \pm 2$, **so** when solving, the points should satisfy both the line and the parabola.

**Wait**, **perhaps** I should approach it differently. Let's use the condition that the line y = mx + c is tangent to the parabola y²=4ax, which has the condition c² = a(1 + m²).

For our parabola y²=8x, a=2, **so** c²=2(1 + m²). For the tangent lines y =x ±2, m=1, c=±2.

**So**, substituting into the parabola equation, we can find the points of contact.

**Alternatively**, using the formula for the point of contact, for the parabola y²=4ax, the point of contact for a tangent y = mx + c is (a/m², 2a/m).

**So**, for our case, a=2, m=1, **so** the point is (2/1, 4/1) = (2,4).

Similarly, for m=-1, the point is (2/1, -4/1) = (2,-4).

Thus, the points of tangency on the parabola are (2,4), (2,-4), (-2,4), (-2,-4). **Wait**, **but** according to the formula, it's (a/m², 2a/m). **So**, if m=1, it's (2,4). If m=-1, it's (2, -4). Similarly, for m=1, it's (2,4), and for m=-1, it's (2,-4). **So**, the four points are (2,4), (2,-4), (-2,4), (-2,-4).

**Wait**, **but** in the Asymptote code, the points are (-1,1), (2,4), (-1,-1), (2,-4). **So**, **perhaps** the points on the circle are (-1,1), (1,-1), (1,1), (-1,-1), and on the parabola, (2,4), (2,-4), (-2,4), (-2,-4). **So**, the quadrilateral is formed by connecting these points in order: (-1,1), (2,4), (-1,-1), (2,-4), and back to (-1,1). That would form a kite-shaped quadrilateral.

**Wait**, **but** the user mentioned four points of tangency, **so perhaps** each tangent touches the circle and the parabola, resulting in four distinct points. **So**, the points are (-1,1), (1,-1), (1,1), (-1,-1) on the circle, and (2,4), (2,-4), (-2,4), (-2,-4) on the parabola.

**So**, the quadrilateral is formed by connecting these points. Let me list them in order: (-1,1), (2,4), (-1,-1), (2,-4), and back to (-1,1). This should form a convex quadrilateral.

Now, to find the area of this quadrilateral, I can use the shoelace formula.

First, list the coordinates in order:

1. (-1,1)
2. (2,4)
3. (-1,-1)
4. (2,-4)
5. (-1,1)  # Closing the polygon

Apply the shoelace formula:

Area = 1/2 |sum over i (x_i y_{i+1} - x_{i+1} y_i)|

Compute each term:

Between (-1,1) and (2,4):

x_i y_{i+1} = (-1)(4) = -4

x_{i+1} y_i = (2)(1) = 2

Term: -4 -2 = -6

Between (2,4) and (-1,-1):

x_i y_{i+1} = (2)(-1) = -2

x_{i+1} y_i = (-1)(4) = -4

Term: -2 - (-4) = 2

Between (-1,-1) and (2,-4):

x_i y_{i+1} = (-1)(-4) = 4

x_{i+1} y_i = (2)(-1) = -2

Term: 4 - (-2) = 6

Between (2,-4) and (-1,1):

x_i y_{i+1} = (2)(1) = 2

x_{i+1} y_i = (-1)(-4) = 4

Term: 2 -4 = -2

Sum all terms: -6 +2 +6 -2 = 0

**Wait**, that can't be right because the area can't be zero. I must have made a mistake in the order of the points or in the calculation.

**Wait**, **perhaps** the order of the points is incorrect. Let me try a different order or check the points.

**Alternatively**, maybe the quadrilateral is actually a rectangle or another shape, **but** the shoelace formula requires the points to be ordered either clockwise or counterclockwise without crossing.

Let me list the points **again**:

(-1,1), (2,4), (-1,-1), (2,-4)

**But** connecting them in this order might cause the polygon to intersect itself. Instead, **perhaps** the correct order is (-1,1), (2,4), (2,-4), (-1,-1), (-1,1). Let me try this order.

Compute the terms:

1. (-1,1) to (2,4):
x_i y_{i+1} = (-1)(4) = -4
x_{i+1} y_i = (2)(1) = 2
Term: -4 -2 = -6

2. (2,4) to (2,-4):
x_i y_{i+1} = (2)(-4) = -8
x_{i+1} y_i = (2)(4) = 8
Term: -8 -8 = -16

3. (2,-4) to (-1,-1):
x_i y_{i+1} = (2)(-1) = -2
x_{i+1} y_i = (-1)(-4) = 4
Term: -2 -4 = -6

4. (-1,-1) to (-1,1):
x_i y_{i+1} = (-1)(1) = -1
x_{i+1} y_i = (-1)(-1) =1
Term: -1 -1 = -2

Sum all terms: -6 -16 -6 -2 = -30

Area = 1/2 | -30 | = 15

**So**, the area is 15.

**Wait**, that seems large, **but** given the points, it might be correct. Let me verify.

**Alternatively**, maybe using vectors or another method would give the same result.

**Alternatively**, since the quadrilateral is symmetric, I can compute the area by finding the area of two triangles and adding them.

**But** the shoelace formula gave me 15, which seems plausible.

**So**, the area of the quadrilateral is 15.

Figure 5: Detail visualization of model outputs on both easy and hard problems.

