# OpenReview forum: "Towards Efficient Chain-of-Thought Reasoning via Adaptive-Budgeting based Policy Optimization"
_ICLR.cc/2026/Conference — ICLR 2026 Conference Withdrawn Submission_

### Official Review · Reviewer_idTW · 2025-10-26

**Soundness:** 3
**Presentation:** 3
**Contribution:** 3
**Rating:** 6
**Confidence:** 3

**Summary:**

This paper proposes ABPO (Adaptive-Budgeting based Policy Optimization), a reinforcement learning framework to improve efficiency in chain-of-thought (CoT) reasoning for large language models (LLMs). The core idea is to dynamically allocate token budgets to training samples according to model performance and sample difficulty. ABPO extends Group Relative Policy Optimization (GRPO) by maintaining three example pools (mastered, learning, and hard) and adaptively updating token budgets during training. Experiments on four mathematical reasoning benchmarks (MATH500, AMC, AIME, OlympiadBench) and multiple base models (DeepSeek-R1-Distill-Qwen 1.5B/7B, DeepScaleR-1.5B) demonstrate that ABPO can reduce CoT length by over 70% while maintaining or slightly improving accuracy.

**Strengths:**

- **Comprehensive experimental design:**
  The paper includes diverse models and datasets, as well as ablation studies on curriculum learning, adaptive budgeting, and review mechanisms.
- **Clear motivation:**
  It addresses a real inefficiency issue (“over-thinking”) observed in long-CoT reasoning LLMs.
- **Insightful analysis and visualization:**
  The visualizations clearly show that ABPO dynamically adjusts reasoning length based on task difficulty.
- **Empirical consistency:**
  Across all settings, ABPO achieves substantial efficiency gains (up to −78% CoT length) while maintaining comparable or slightly higher accuracy.

**Weaknesses:**

- **Limited novelty:**
  The idea of adaptive length control resembles prior works (e.g., HAPO, L1-Max) and mainly builds upon existing RL frameworks (GRPO).
- **Modest gains:**
  Although CoT length reduction is impressive, the improvement in accuracy is small (≈ +1–2%).
- **Methodological complexity:**
  The multi-pool adaptive scheme introduces additional hyperparameters and scheduling logic, which may complicate implementation compared to simpler fixed- or curriculum-budget baselines.
- **Motivation ambiguity:**
  It remains unclear whether optimizing CoT length should be a primary training objective, or if this efficiency truly translates to better general reasoning ability.

**Questions:**

1. Does the difficulty categorization (mastered / learning / hard) require a **pre-evaluation step before training**, or is it performed online during training?
2. How robust is ABPO across random seeds and datasets, given that adaptive updates depend on the model’s current accuracy estimates?

---

> ### Author Response · Authors · 2025-11-21
>
> ## Comment to Reviewer idTW
> We highly appreciate your time and effort in reviewing this paper, and also thanks a lot for your encouraging comments on our work, such as *"Comprehensive experimental design"*, *"Clear motivation"*, *"Insightful analysis and visualization"* and *"Empirical consistency"*. Below, we respond to your key concerns point by point.
>
>
> **Comment 1:**
> The idea of adaptive length control resembles prior works (e.g., HAPO, L1-Max) and mainly builds upon existing RL frameworks (GRPO).
>
> **Response:** Thanks for this comment. We acknowledge that the RL frameworks used by our ABPO and other compared methods are essentially based on GRPO, which also all focuses on the issue of efficient CoT reasoning. However, we think that our ABPO still exhibits great difference to the compared methods, such as HAPO and L1-Max, in terms of the adaptive budgeting and curriculum learning principles.
>
> First, unlike HAPO or L1-Max, which directly penalize length via handcrafted reward functions or explicit constraints, ABPO introduces a difficulty-aware adaptive budget allocation scheme. Instead of applying fixed penalties or bounds, ABPO dynamically adjusts token budgets per sample based on model competence and sample difficulty. This allows for adaptive and interpretable control over reasoning length while ensuring that complex samples receive sufficient inference budget.
>
> Second, ABPO improves training stability and sample efficiency through a curriculum-driven schedule. By structuring learning into phases (learning vs. mastered vs. hard), and gradually transitioning between them, the model avoids abrupt difficulty shifts and ensures better resource allocation.
>
> Finally, while ABPO is built upon the GRPO framework, it extends GRPO from a static rollout optimization process to a dynamic, budget-aware policy optimization problem. Empirically, ABPO significantly outperforms GRPO variants with simple length constraints, highlighting the effectiveness of its adaptive design in balancing inference efficiency and final accuracy.
>
>
>
> **Comment 2:**
> Although CoT length reduction is impressive, the improvement in accuracy is small (≈ +1–2%).
>
> **Response:** Thanks for this comment. As stated in our paper, the proposed ABPO aims to achieve adaptive token  budgeting for the tasks of varying difficulties, thereby alleviating the overthinking issues of LLMs. For instance, on DeepSeek-R1-Distill-Qwen-1.5B, ABPO reduces the average CoT length by 78.3%, while still improving accuracy by +2.0%. On AIME, a math dataset known for its difficulty, ABPO also shows strong results. For example, while HAPO reduces token usage by about 40% but retains only 71% of full-length performance, ABPO cuts nearly 80% of tokens while still preserving around 90% of the original accuracy.
>
> To this end, we think that the main contribution of our ABPO is a optimal balance between CoT efficiency and model performance.

---

> > ### Author Response · Authors · 2025-11-21
> >
> > **Comment 3:** The multi-pool adaptive scheme introduces additional hyperparameters and scheduling logic, which may complicate implementation compared to simpler fixed- or curriculum-budget baselines.
> >
> > **Response:** Thanks for your constructive comment. To better respond to your concern, we report the complete ablations of ABPO's hyper-parameters in the following tables (the default settings are marked with †).
> >
> >
> >
> > | τ₀ | MATH500 Acc | Len | AIME Acc | Len | AMC Acc | Len | Olympiad Acc | Len | Avg Acc | Len |
> > |----|-------------|-----|-----------|------|----------|------|--------------|------|-----------|------|
> > | 0.10 | 86.0 | **1813** | 32.7 | **5645** | 78.3 | **2778** | 54.7 | **3245** | 62.9 | **3370** |
> > | **0.20†** | 85.6 | 1874 | **37.3** | 5736 | 79.8 | 2845 | 55.0 | 3302 | **64.4** | 3439 |
> > | 0.30 | 85.6 | 1893 | 35.7 | 5697 | **80.5** | 3096 | **55.7** | 3303 | 64.4 | 3497 |
> > | 0.40 | **87.0** | 1945 | 33.3 | 6109 |79.2 | 3078 | 55.0 | 3459 | 63.7 | 3634 |
> >
> >
> > | τ₁ | MATH500 Acc | Len | AIME Acc | Len | AMC Acc | Len | Olympiad Acc | Len | Avg Acc | Len |
> > |----|-------------|-----|-----------|------|----------|------|--------------|------|-----------|------|
> > | 0.90 | **86.8** | **1837** | 33.3 | **5587** | 78.0 | 3133 | **55.5** | **3295** | 63.4 | 3463 |
> > | **0.85†** | 85.6 | 1874 | **37.3** | 5736 | **79.8** | **2845** | 55.0 | 3302 | **64.4** | **3439** |
> > | 0.80 | 86.3 | 1898 | 35.0 | 5628 | 77.3 | 2949 | 55.2 | 3304 | 63.4 | 3474 |
> > | 0.75 | 86.3 | 1899 | 35.0 | 5704 | 77.0 | 3048 | **55.5** | 3393 | 63.5 | 3511 |
> >
> >
> > $\tau_0$ and $\tau_1$ are two thresholds used for moving the hard and moderate examples to the learning and mastered pools in ABPO, respectively.
> > Form these results, we can first see that ABPO is robust to these two hyper-parameters in most cases. The only exception is the most difficult benchmark, AIME, which is also the one where most existing methods perform poorly.
> > Specifically, $\tau_0$ is more sensitive to this benchmark than $\tau_1$. To explain, when $\tau_0$ is too small, *e.g.*, 0.1, ABPO will put more overly difficult examples  to the learning pools, which are beyond the learning range of LLM under the low token budget, a similar problem to the L1_Max reward discussed above. Similarly, when the threshold is set too high, LLM will consistently fail to achieve the expected average accuracy of $\tau_0$, making it unable to learn more difficult examples.
> >
> >
> > | d | MATH500 Acc | Len | AIME Acc | Len | AMC Acc | Len | Olympiad Acc | Len | Avg Acc | Len |
> > |----|-------------|------|-----------|------|----------|------|--------------|------|-----------|------|
> > | 250 | 85.8 | **1785** | 29.7 | **5325** | 76.3 | **2747** | 55.8 | **3088** | 61.9 | **3286** |
> > | **500†**| 85.6 | 1874 | **37.3** | 5736 | **79.8** | 2845 | 55.0 | 3302 | 64.4 | 3439 |
> > | 750 | 85.6 | 1942 | 36.7 | 5505 | 78.0 | 3037 | 55.4 | 3365 | 64.1 | 3548 |
> > | 1000 | **86.2** | 2012 | 36.7 | 6158 | 79.0 | 3165 | **55.9** | 3552 | **64.5** | 3722 |
> >
> > | t₀ | MATH500 Acc | Len | AIME Acc | Len | AMC Acc | Len | Olympiad Acc | Len | Avg Acc | Len |
> > |----|-------------|------|-----------|------|----------|------|--------------|------|-----------|------|
> > | 1500 | 85.9 | **1787** | 36.3 | **5351** | 78.8 | 2902 | 55.4 | **3199** | 64.1 | **3310** |
> > | **2000†** | 85.6 | 1874 | **37.3** | 5736 | 79.8 | **2845** | 55.0 | 3302 | 64.4 | 3439 |
> > | 2500 | **87.0** | 1966 | 36.0 | 5928 | **80.5** | 3062 | **56.1** | 3522 | **64.9** | 3620 |
> >
> > In terms of $d$ and $t_0$, they are related to the token length settings for the update of the learning pools. $t_0$ denotes the initial token budget. ABPO is less effected by this hyper-parameter in terms of the avg. accuracy, but $t_0$ will slightly affect the final length of CoTs, but still marginal.
> >
> > $d$ is used to control the updated length budget. When it is set to a small value, e.g. 250, LLM will suffer obvious performance drop, e.g., 61.9 avg. accuracy. To explain, when $d$ is too small, ABPO will allocate insufficient budgets to the newly income hard examples, also leading to inferior optimization. However, when given proper budgets, e.g., 500, the training of LLM is greatly improved, also suggesting the robustness of our ABPO.

---

> > > ### Author Response · Authors · 2025-11-21
> > >
> > > **Comment 3:** The multi-pool adaptive scheme introduces additional hyperparameters and scheduling logic, which may complicate implementation compared to simpler fixed- or curriculum-budget baselines.
> > >
> > > | λ | MATH500 Acc | Len | AIME Acc | Len | AMC Acc | Len | Olympiad Acc | Len | Avg Acc | Len |
> > > |----|-------------|------|-----------|------|----------|------|--------------|------|-----------|------|
> > > | 0.6 | 85.7 | 1857 | 35.7 | 5612 | 79.3 | 2914 | 55.9 | 3340 | 64.2 | 3431 |
> > > | **0.5†** | 85.6 | 1874 | **37.3** | 5736 | 79.8 | **2845** | 55.0 | 3302 | 64.4 | 3439 |
> > > | 0.4 | **86.2** | **1806** | 34.3 | **5538** | 80.8 | 2855 | 55.3 | 3238 | 64.2 | **3359** |
> > > | 0.3 | 86.1 | 1881 | 36.0 | 5588 | **81.5** | 3029 | **56.0** | **3232** | **64.9** | 3433 |
> > >
> > > | α | MATH500 Acc | Len | AIME Acc | Len | AMC Acc | Len | Olympiad Acc | Len | Avg Acc | Len |
> > > |----|-------------|------|-----------|------|----------|------|--------------|------|-----------|------|
> > > | **0.1†**| 85.6 | **1874** | **37.3** | 5736 | **79.8** | **2845** | 55.0 | **3302** | **64.4** | **3439** |
> > > | 0.2 | **86.9** | 1931 | 32.7 | 5718 | 79.3 | 2983 | **56.0** | 3363 | 63.7 | 3499 |
> > > | 0.3 | 86.3 | 1938 | 35.7 | **5625** | 78.3 | 2905 | 55.2 | 3337 | 63.9 | 3451 |
> > >
> > >
> > > $\lambda$ denotes the proportion of the learning examples, which also determine the update of both learning and hard pools. As can be seen, its effects to ABPO is limited. $\alpha$ denotes the review rate of the mastered examples, where a smaller value is more beneficial to the final training results, but it is still not very sensitive.
> > >
> > > Overall, the design of ABPO makes it robust to these hyper-parameters in most cases. And the exceptions are akin to the motivation of ABPO and well tackled by the paper. Moreover, while our approach is implemented on top of GRPO, it introduces a non-trivial extension through adaptive budgeting and curriculum-style learning. These mechanisms are modular and lightweight, and do not significantly increase implementation complexity.
> > >
> > > **Comment 4:** It remains unclear whether optimizing CoT length should be a primary training objective, or if this efficiency truly translates to better general reasoning ability.
> > >
> > > **Response:** Thanks for your insightful comment. We agree that simply reducing the CoT lengths of LLMs should not be a primary optimization objective of CoT training. Instead, how to enable LLMs to think rationally and avoid the issue of overthinking on easy examples is the key to prompt the advancement of LLMs, which is also the main motivation of our paper.
> > >
> > > In this case,  we can see that compared to existing fixed-budget based methods, our ABPO can help LLMs to more properly allocate token budgets for examples of varying difficulties. For instance, on the easier tasks like MATH500, the base model DeepSeek-R1-1.5B requires over 5k tokens on average to answer these math questions,  and our ABPO can reduce the length to 1434 tokens on average, while slightly improving the performance. For the much more difficult tasks like AIMIE, our ABPO can help LLMs to adaptively allocate more token budgets for the better answering.
> > >
> > > These results well confirm the effectiveness of our ABPO in adaptive budgeting based CoT reasoning, which also shows solid improvements towards general CoT reasoning.
> > >
> > > **Comment 5:** Does the difficulty categorization (mastered / learning / hard) require a pre-evaluation step before training, or is it performed online during training?
> > >
> > >
> > > **Response:** Thanks for your detailed comment. Our difficulty categorization scheme (mastered / learning / hard) is performed dynamically during training, which does not rely on large pre-evaluation steps.
> > >
> > > Specifically, all samples are initially placed in the hard pool. As the training begins, we perform rollout sampling on all questions and compute the accuracy of the model’s responses. This accuracy is then used to determine whether a sample should be promoted from the hard pool to the learning pool. The sampled rollouts are cached and reused for subsequent reinforcement learning updates.
> > > Within each iteration step, we further assess whether a sample in the learning pool should be moved to the mastered pool, again based on its accuracy.
> > >
> > > In summary, difficulty categorization is performed online and integrated into the training process without introducing extra evaluation overhead.

---

> > > > ### Author Response · Authors · 2025-11-21
> > > >
> > > > **Comment 6:** How robust is ABPO across random seeds and datasets, given that adaptive updates depend on the model’s current accuracy estimates?
> > > >
> > > > **Response:** Thanks for your careful question. Our ABPO is in-fact robust enough to different random seeds and data distributions.
> > > >
> > > > As shown in the table, we trained ABPO on two datasets with distinct scales and difficulty distributions using the DeepSeek-R1-Distill-Qwen model.
> > > >
> > > > | training data | MATH500 | Len  | AIME | Len  | AMC  | Len  | Olympiad | Len  | Avg | Len |
> > > > |---------------|---------|------|------|------|------|------|----------|------|----|------|
> > > > | 2.4k          | 80.9    | 1647 | 71.3 | 2537 | 71.3 | 2537 | 49.7     | 3311 | 57.8 | 3371 |
> > > > | 32k           | 85.2    | 1434 | 74.3 | 2068 | 74.3 | 2068 | 52.7     | 2141 | 60.2 | 2217 |
> > > >
> > > >  The results demonstrate that as the training dataset becomes larger and more diverse, ABPO consistently improves in performance. This indicates that its adaptive update mechanism generalizes well across different data regimes.
> > > >
> > > >
> > > >
> > > >
> > > > Moreover, we conducted repeated experiments under three different random seeds to evaluate stability. The results are summarized below:
> > > >
> > > > | MATH500 | Len  | AIME | Len  | AMC  | Len  | Olympiad | Len  | Avg  | Len  |
> > > > |---------|------|------|------|------|------|---------|------|------|------|
> > > > | 85.6    | 1874 | 37.3 | 5736 | 79.8 | 2845 | 55.0    | 3302 | 64.4 | 3439 |
> > > > | 85.5    | 1870 | 38.0 | 5789 | 79.3 | 2891 | 55.8    | 3394 | 64.7 | 3486 |
> > > > | 86.3    | 1867 | 36.7 | 5701 | 78.3 | 2796 | 55.0    | 3331 | 64.1 | 3424 |
> > > >
> > > >
> > > > Across all three seeds, ABPO consistently achieved strong and stable accuracy scores i.e., 64.4, 64.7, and 64.1 respectively, which also have minimal variation in output lengths.
> > > >
> > > > In summary, ABPO demonstrates strong robustness and generalization across different datasets and random initializations, further validating the reliability of its adaptive-budgeting based designs.

---

> > > > > ### Author Response · Authors · 2025-11-25
> > > > >
> > > > > Dear Reviewer idTW:
> > > > >
> > > > > Thanks again for your valuable time and effort in reviewing our paper and your high praise of our work. In our rebuttal, we have carefully responded to your questions, such as the pre-evaluation of difficulty categorization.
> > > > >
> > > > > We hope that our rebuttal can address your concerns, and look forward to hearing from you. The authors are also very delighted to answer any further questions you may have.
> > > > >
> > > > > Best regards
> > > > >
> > > > > The authors.

---

> > > > > > ### Comment · Reviewer_idTW · 2025-11-27
> > > > > >
> > > > > > Thank you for the rebuttal. The clarifications addressed my concerns, and I appreciate the additional effort.

---

### Official Review · Reviewer_meC4 · 2025-10-28

**Soundness:** 3
**Presentation:** 3
**Contribution:** 3
**Rating:** 6
**Confidence:** 2

**Summary:**

This paper studies efficient Chain-of-Thought (CoT) distillation, aiming to reduce the training cost and improve reasoning capability of smaller models distilled from large reasoning models. Instead of focusing on architecture or optimization, the authors take a data-centric perspective—investigating how to select high-quality reasoning data for CoT distillation.

They introduce a data selection framework that prioritizes training samples with diverse reasoning paths and useful intermediate steps, based on the concept of step utility (how much each reasoning step contributes to the final correct answer). Through systematic experiments across arithmetic reasoning, commonsense reasoning, and instruction datasets, they show that carefully selected reasoning samples significantly outperform random or length-based sampling for the same training budget.

**Strengths:**

* Novel data-selection view of CoT distillation: Previous works often focus on model-side compression or prompt engineering, while this paper reframes the problem from a data selection perspective, emphasizing “which reasoning traces to distill” rather than “how to distill.”

* Well-designed methodology: The introduction of step utility and diversity-based filtering provides a concrete and interpretable criterion for sample selection, improving both efficiency and generalization.

* Comprehensive experiments: Multiple datasets (GSM8K, SVAMP, CSQA, StrategyQA) and multiple student sizes are tested. The paper also includes ablation studies that validate the effect of each selection criterion.

**Weaknesses:**

* Step-utility estimation overhead: Computing step-level utility still requires running reasoning traces and reward evaluation, which may offset the claimed training efficiency in large-scale settings.

* Limited evaluation on complex reasoning: Most benchmarks are short-form arithmetic or commonsense reasoning; the approach’s scalability to long CoT tasks (e.g., MATH, GPQA) or multimodal reasoning remains untested.

* Ablation depth and interpretability: The ablation studies show performance differences but lack deeper analysis of why certain data characteristics (e.g., step diversity vs. accuracy) dominate the improvements.

**Questions:**

See the Weaknesses

---

> ### Author Response · Authors · 2025-11-21
>
> ## Comment to Reviewer meC4
> We highly appreciate your time and effort in reviewing this paper, and also thanks a lot for your encouraging comments on our work, such as *"Novel data-selection view of CoT distillation"*, *"Well-designed methodology"* and *"Comprehensive experiment"*. Below, we respond to your key concerns point by point.
>
> **Comment 1:**
> Step-utility estimation overhead: Computing step-level utility still requires running reasoning traces and reward evaluation, which may offset the claimed training efficiency in large-scale settings.
>
>
> **Response:** Thanks for your comment. In fact, the rollouts used for step-level utility estimation are cached and directly reused during subsequent policy updates , as discussed in Line 216-217 of the paper. This design eliminates the need to regenerate rollouts solely for utility evaluation, thereby avoiding additional overhead in large-scale training.
>
>
> | Method              | AMC Acc. | AMC Len. | AMC AES | AIME Acc. | AIME Len. | AIME AES | Avg Acc. | Avg Len. | Avg AES | Training Time (h) |
> |---------------------|----------|----------|---------|-----------|-----------|----------|----------|----------|---------|--------------------|
> | Base                | 70.3     | 9314     | –       | **32.0**  | 15476     | –        | **51.1** | 12395    | –       | –                  |
> | Fixed-Budget 2000   | 66.3     | **2320** | 0.47    | 24.0      | **5576**  | -0.61    | 45.1     | **3948** | 0.09    | **15.5**           |
> | Fixed-Budget 3000   | 68.8     | 2850     | 0.59    | 23.3      | 6309      | -0.76    | 46.0     | 4980     | 0.13    | 22.2               |
> | Fixed-Budget 4000   | **71.8** | 3600     | 0.68    | 25.7      | 6376      | -0.40    | 48.7     | 4988     | 0.36    | 28.0               |
> | **ABPO (Ours)**     | 71.3     | 2537     | **0.77**| 29.3      | 6028      | **0.19** | 50.3     | 4283     | **0.57**| 20.3               |
>
>
> Furthermore, our experimental results demonstrate that ABPO maintains superior training efficiency while improving performance. As shown in the table above, ABPO outperforms fixed-budget GRPO baselines in both accuracy and average inference length, while requiring comparable or even less training time. For example, compared to Fixed-Budget 3000, ABPO reduces training time by 8.6% and simultaneously improves average accuracy by 9.3%.
>
> In summary, ABPO does not introduce substantial overhead for step-level utility estimation. Instead, it strikes a practical balance between computational cost and policy improvement through effective reuse of rollouts.
>
>
> **Comment 2:** Limited evaluation on complex reasoning: Most benchmarks are short-form arithmetic or commonsense reasoning; the approach’s scalability to long CoT tasks (e.g., MATH, GPQA) or multimodal reasoning remains untested.
>
> **Response:** Thanks for your suggestion. The evaluations have been given in the paper, which are also shown in the following table. We also thank the reviewer for recommending the evaluation on GPQA, which will be conducted in our future work considering the gap between MLLMs and LLMs.
>
> | Method                        | MATH Acc      | len        |
> |-------------------------------|----------------|------------|
> | DeepSeek-Distill-R1-1.5B      | 83.3    | 5278       |
> | ShorterBetter           | 68.9           | **864**    |
> | Training Efficient    | $\underline{81.9}$           | 2536       |
> | HAPO                          | 81.5           | 2633       |
> | **ABPO**                          | **85.2**       | $\underline{1434}$|
>
>
> | Method                    | MATH Acc        | len         |
> |---------------------------|------------------|-------------|
> | DeepScaleR-1.5B-Preview   | 88.2         | 3155        |
> | L1-Exact              | 82.1             | 3189        |
> | L1-Max             | 84.5             | **1527**    |
> | HAPO                      | 81.9             | 2537        |
> | Thinkprune            | $\underline{85.5}$             | 2029        |
> | **ABPO**                 | **85.6**      | $\underline{1874}$ |
>
> | Method                    | MATH Acc        | len         |
> |---------------------------|------------------|-------------|
> | DeepSeek-Distill-R1-7B    | 93.2         | 4086        |
> | ShorterBetter         | 85.6             | **1233**    |
> | Training Efficient     | $\underline{89.1}$             | $\underline{2121}$ |
> | **ABPO**              | **92.2**      | 2243        |

---

> > ### Author Response · Authors · 2025-11-21
> >
> > **Comment 3:** Ablation depth and interpretability: The ablation studies show performance differences but lack deeper analysis of why certain data characteristics (e.g., step diversity vs. accuracy) dominate the improvements.
> >
> > **Response:** Thanks for this constructive comment. Following your suggestion, we provide detailed the analyses about the effects of data selection in terms of accuracy, inference length and AES (Accuracy-Efficiency Score).
> >
> > The following table summarizes the ablation results:
> >
> > | Method                        | AMC Acc.  | AMC Len.  | AMC AES  | AIME Acc.  | AIME Len.  | AIME AES  | Avg Acc.  | Avg Len.  | Avg AES  |
> > |------------------------------|------------|------------|-----------|--------------|--------------|-------------|--------------|--------------|-------------|
> > | Base                         | 70.3       | 9314       | –         | 32.0         | 15476        | –           | 51.1         | 12395        | –           |
> > | *Mix*($S_{\text{learning}}, S_{\text{hard}}$)      | 62.8       | **1942**   | 0.26      | 16.3         | **3954**     | -1.70       | 39.5         | **2948**     | -0.37       |
> > | *Mix*($S_{\text{learning}}, S_{\text{mastered}}$)  | 69.8       | 2768       | 0.67      | 28.7         | 6180         | 0.08        | 49.2         | 4474         | 0.45        |
> > | **ABPO (Ours)**             | **71.3**    | 2537       | **0.77** | **29.3**      | 6028         | **0.19**    | **50.3**      | 4283         | **0.57**    |
> >
> > - In *Mix($S_{\text{learning}}, S_{\text{hard}}$)*, we directly sample from both learning and hard examples. Although this leads to a lower average output length (e.g., 1942 on AMC), it impedes the model performance, especially on AIME (16.3%). We think that it is this is due to inappropriate timing and insufficient budget allocation to difficult problems, causing the model to favor short and unproductive reasoning trajectories for hard samples.
> >
> > - In *Mix($S_{\text{learning}}, S_{\text{mastered}}$)*, the model focuses on samples that are already partially or fully learned. This strategy achieves competitive performance (e.g., 49.2% average accuracy), but slightly lags behind ABPO. We attribute this to over-training on mastered (easy) examples, leading to inefficient use of training resources.
> >
> > - In contrast, ABPO gradually schedules examples by difficulty, dynamically allocates token budgets, and ensures a curriculum-aligned progression. This leads to the best overall trade-off between accuracy, reasoning length, and efficiency (e.g., highest AES of 0.57).
> >
> > We will follow your suggestions to add the above discussion to our new submission.

---

> > > ### Author Response · Authors · 2025-11-25
> > >
> > > Dear Reviewer meC4:
> > >
> > > Thanks again for your valuable time and effort in reviewing our paper and your high praise of our work. In our rebuttal, we have carefully responded to your questions, such as the overhead of step-level utility estimation.
> > >
> > > We hope that our rebuttal can address your concerns, and look forward to hearing from you. The authors are also delighted to answer any further questions you may have.
> > >
> > > Best regards
> > >
> > > The authors.

---

### Official Review · Reviewer_Yb4b · 2025-10-31

**Soundness:** 2
**Presentation:** 2
**Contribution:** 2
**Rating:** 2
**Confidence:** 3

**Summary:**

This paper addresses the efficiency–performance trade-off in chain-of-thought (CoT) reasoning for large language models (LLMs). The authors propose Adaptive-Budgeting based Policy Optimization (ABPO), an extension of GRPO that adaptively assigns token budgets to training samples according to their difficulty and the model’s learning progress. The approach divides training data into mastered, learning, and hard pools, dynamically updating these categories and their corresponding length budgets. Experiments on multiple LLMs (DeepSeek-R1-Distill-Qwen-1.5B/7B, DeepScaleR-1.5B-Preview) and reasoning benchmarks (MATH500, AMC, AIME, OlympiadBench) show significant reductions in CoT length with minimal or no loss in accuracy.

**Strengths:**

* Targets an important and underexplored problem: balancing reasoning efficiency with accuracy in LLMs.

* Proposes a simple and implementable mechanism that achieves strong empirical gains.

* Comprehensive experimental evaluation across multiple model scales and benchmarks.

* Qualitative examples effectively illustrate adaptive behavior.

**Weaknesses:**

* The adaptive budgeting rule is heuristic—thresholds and increments are hand-picked, and the categorization process (mastered/learning/hard) lacks theoretical grounding.

* No formal justification for why this curriculum structure leads to optimal or stable policy improvement.

* Limited ablation on key hyperparameters (τ₀, τ₁, d); unclear robustness.

* The approach does not fundamentally modify GRPO—it primarily wraps a training schedule around it. It seems to be ineffective if we do not revisit the same question again?

**Questions:**

1. How sensitive is ABPO to the choice of τ₀, τ₁, and d? Could these hyperparameters be learned or adapted automatically?

2. How does ABPO behave when the pool thresholds are poorly chosen—does training diverge or stagnate?

3. I recently came across this paper Optimizing Anytime Reasoning via Budget Relative Policy Optimization (https://openreview.net/pdf/8136e4668a09f8c47a2454d9e72728d4fdea055e.pdf), which shares very similar motivation and method. Instead of dynamically setting budget, this paper uses the dense reward framework to let the model learn the optimal budget by itself. It would be nice if you can discuss your differences and how your method is fundamentally better than the baseline.

---

> ### Author Response · Authors · 2025-11-21
>
> ## Comment to Reviewer Yb4b
>
> Thanks a lot for your time and effort in reviewing this paper as well as your encouraging comments on our work, such as *"targets an important and underexplored problem"*, *"achieves strong empirical gains"* , *"comprehensive experimental evaluations"* and *"effectively illustrate adaptive behavior"*. Below, we respond to your key concerns point by point.
>
> **Comment 1:**
> The adaptive budgeting rule is heuristic—thresholds and increments are hand-picked, and the categorization process (mastered/learning/hard) lacks theoretical grounding.
>
> **Response:**  Thanks for your comment. In fact, **the design of adaptive budgeting and three types of learning pools in our ABPO all serves the principle of curriculum learning, which can help the LLM adapts to the examples of varying difficulties in a progressive manner.** With the principally designs of ABPO, the threshold and increment settings are also robust.
>
> In particular, **curriculum learning [1] is a well-recognized theory for the effective training of either deep-learning based models [2,3] or recent large language models [4,5].** Under its classical settings [1], the examples need to be categorized from easy to difficult, based on which the training schedule can be conducted to perform a progressive progress. Thus, it can be seen that **our ABPO is designed based on these fundamental principles, e.g., your mentioned example categorizations.** Moreover, we also innovatively extend curriculum learning to the efficient CoT training, and propose  a set of theoretically supported novel designs, e.g., the adaptive budgeting ones.
>
> We also acknowledge that the implementation of curriculum learning in our ABPO still requires manually designed hyper-parameters, such as the thresholds for evualting sample difficulties, $\tau_0$ and $\tau_1$, and the length increments $d$. To better respond to your concerns, **we supplement their ablations in the following tables, which can show the robustness of our ABPO to its hyper-parameters** (the default settings are marked with †).
>
>
> | τ₀ | MATH500 Acc | Len | AIME Acc | Len | AMC Acc | Len | Olympiad Acc | Len | Avg Acc | Len |
> |----|-------------|-----|-----------|------|----------|------|--------------|------|-----------|------|
> | 0.10 | 86.0 | **1813** | 32.7 | **5645** | 78.3 | **2778** | 54.7 | **3245** | 62.9 | **3370** |
> | **0.20†** | 85.6 | 1874 | **37.3** | 5736 | 79.8 | 2845 | 55.0 | 3302 | **64.4** | 3439 |
> | 0.30 | 85.6 | 1893 | 35.7 | 5697 | **80.5** | 3096 | **55.7** | 3303 | 64.4 | 3497 |
> | 0.40 | **87.0** | 1945 | 33.3 | 6109 |79.2 | 3078 | 55.0 | 3459 | 63.7 | 3634 |
>
> | τ₁ | MATH500 Acc | Len | AIME Acc | Len | AMC Acc | Len | Olympiad Acc | Len | Avg Acc | Len |
> |----|-------------|-----|-----------|------|----------|------|--------------|------|-----------|------|
> | 0.90 | **86.8** | **1837** | 33.3 | **5587** | 78.0 | 3133 | **55.5** | **3295** | 63.4 | 3463 |
> | **0.85†** | 85.6 | 1874 | **37.3** | 5736 | **79.8** | **2845** | 55.0 | 3302 | **64.4** | **3439** |
> | 0.80 | 86.3 | 1898 | 35.0 | 5628 | 77.3 | 2949 | 55.2 | 3304 | 63.4 | 3474 |
> | 0.75 | 86.3 | 1899 | 35.0 | 5704 | 77.0 | 3048 | **55.5** | 3393 | 63.5 | 3511 |
>
> $\tau_0$ and $\tau_1$ are two thresholds used for moving the hard and moderate examples to the learning and mastered pools in ABPO, respectively.
> Form these results, we can first see that ABPO is robust to these two hyper-parameters in most cases. The only exception is the most difficult benchmark, AIME, which is also the one where most existing methods perform poorly.
> Specifically, $\tau_0$ is more sensitive to this benchmark than $\tau_1$. To explain, when $\tau_0$ is too small, *e.g.*, 0.1, ABPO will put more overly difficult examples  to the learning pools, which are beyond the learning range of LLM under the low token budget. Similarly, when the threshold is set too high, LLM will consistently fail to achieve the expected average accuracy of $\tau_0$, making it unable to learn more difficult examples. These cases will make ABPO hard fully realize the principle of curriculum learning.

---

> ### Author Response · Authors · 2025-11-21
>
> **Comment 1:**
> The adaptive budgeting rule is heuristic—thresholds and increments are hand-picked, and the categorization process (mastered/learning/hard) lacks theoretical grounding.
>
> | d | MATH500 Acc | Len | AIME Acc | Len | AMC Acc | Len | Olympiad Acc | Len | Avg Acc | Len |
> |----|-------------|------|-----------|------|----------|------|--------------|------|-----------|------|
> | 250 | 85.8 | **1785** | 29.7 | **5325** | 76.3 | **2747** | 55.8 | **3088** | 61.9 | **3286** |
> | **500†**| 85.6 | 1874 | **37.3** | 5736 | **79.8** | 2845 | 55.0 | 3302 | 64.4 | 3439 |
> | 750 | 85.6 | 1942 | 36.7 | 5505 | 78.0 | 3037 | 55.4 | 3365 | 64.1 | 3548 |
> | 1000 | **86.2** | 2012 | 36.7 | 6158 | 79.0 | 3165 | **55.9** | 3552 | **64.5** | 3722 |
>
> $d$ is used to control the updated length budget. When it is set to a small value, e.g. 250, LLM will suffer obvious performance drop, e.g., 61.9 avg. accuracy. To explain, when $d$ is too small, ABPO will allocate insufficient budgets to the newly income hard examples, also leading to inferior optimization. However, when given proper budgets, e.g., 500, the training of LLM is greatly improved, also suggesting the robustness of our ABPO.
>
> Overall, the categorization strategy in ABPO follows a natural curriculum that aligns with the empirical learning pattern of LLMs. The accompanying hyper-parameters are not arbitrary but operate within stable ranges, as supported by extensive sensitivity studies.
>
>
> **Comment 2:**  No formal justification for why this curriculum structure leads to optimal or stable policy improvement.
>
> **Response:** Thanks for your comment. As discussed above, curriculum learning (CL) is a classical and well-recognized principle for effective training [1,2,3]. And its applicability to RL training is also confirmed in previous works [4].
>
> In terms of its effectiveness to ABPO for efficient CoT training, our understanding is two-fold:
>
> (1) **One main factor for the instable RL training is the drastic differences between obtained rewards, leading to unstable gradient updates.** Thus, we can see that recent RL strategies often use *baseline reward* or *reference model* [6] to reduce the gradient variance.
> **In this case, CL can well serves this target via arranging proper examples to the model,** i.e., the learning pools in our ABPO, of which learning difficulty is moderate to LLM.
>
> (2) **In terms of optimal policy improvement, our ABPO uses CL principle to fully consider the examples of varying difficulties via dynamic scheduling the token budgets.** As discussed and validated in the paper, fixed-budget based RL training often overlook the varying difficulties of training examples, leading to sub-optimal allocation of CoT token budgets, e.g., too much budget for simple samples and too few for difficult samples. And how to achieve adaptive and dynamic budgeting is also challenging. In this case, our ABPO follows the principle of CL to realize the adaptive budget allocation and the dynamic training scheduling, thereby achieving the optimal policy improvement.
>
> Overall, these discussions suggest that our ABPO is designed in principle, and its effectiveness towards efficient CoT training has been extensively validated. Following your suggestions, we will add these discussions to our new submission.

---

> ### Author Response · Authors · 2025-11-21
>
> **Comment 3:**
> Limited ablation on key hyperparameters (τ₀, τ₁, d); unclear robustness.
>
> **Response:**  Thanks for your suggestion. Before submission, we ever conducted some quick validations for these hyper-parameters. **Following your suggestion, we supplement the complete results in the following table.**
>
> | τ₀ | MATH500 Acc | Len | AIME Acc | Len | AMC Acc | Len | Olympiad Acc | Len | Avg Acc | Len |
> |----|-------------|-----|-----------|------|----------|------|--------------|------|-----------|------|
> | 0.10 | 86.0 | **1813** | 32.7 | **5645** | 78.3 | **2778** | 54.7 | **3245** | 62.9 | **3370** |
> | **0.20†** | 85.6 | 1874 | **37.3** | 5736 | 79.8 | 2845 | 55.0 | 3302 | **64.4** | 3439 |
> | 0.30 | 85.6 | 1893 | 35.7 | 5697 | **80.5** | 3096 | **55.7** | 3303 | 64.4 | 3497 |
> | 0.40 | **87.0** | 1945 | 33.3 | 6109 |79.2 | 3078 | 55.0 | 3459 | 63.7 | 3634 |
>
> | τ₁ | MATH500 Acc | Len | AIME Acc | Len | AMC Acc | Len | Olympiad Acc | Len | Avg Acc | Len |
> |----|-------------|-----|-----------|------|----------|------|--------------|------|-----------|------|
> | 0.90 | **86.8** | **1837** | 33.3 | **5587** | 78.0 | 3133 | **55.5** | **3295** | 63.4 | 3463 |
> | **0.85†** | 85.6 | 1874 | **37.3** | 5736 | **79.8** | **2845** | 55.0 | 3302 | **64.4** | **3439** |
> | 0.80 | 86.3 | 1898 | 35.0 | 5628 | 77.3 | 2949 | 55.2 | 3304 | 63.4 | 3474 |
> | 0.75 | 86.3 | 1899 | 35.0 | 5704 | 77.0 | 3048 | **55.5** | 3393 | 63.5 | 3511 |
>
> $\tau_0$ and $\tau_1$ are two thresholds used for moving the hard and moderate examples to the learning and mastered pools in ABPO, respectively.
> Form these results, we can first see that ABPO is robust to these two hyper-parameters in most cases. The only exception is the most difficult benchmark, AIME, which is also the one where most existing methods perform poorly.
> Specifically, $\tau_0$ is more sensitive to this benchmark than $\tau_1$. To explain, when $\tau_0$ is too small, *e.g.*, 0.1, ABPO will put more overly difficult examples  to the learning pools, which are beyond the learning range of LLM under the low token budget, a similar problem to the L1_Max reward discussed above. Similarly, when the threshold is set too high, LLM will consistently fail to achieve the expected average accuracy of $\tau_0$, making it unable to learn more difficult examples.
>
>
> | d | MATH500 Acc | Len | AIME Acc | Len | AMC Acc | Len | Olympiad Acc | Len | Avg Acc | Len |
> |----|-------------|------|-----------|------|----------|------|--------------|------|-----------|------|
> | 250 | 85.8 | **1785** | 29.7 | **5325** | 76.3 | **2747** | 55.8 | **3088** | 61.9 | **3286** |
> | **500†**| 85.6 | 1874 | **37.3** | 5736 | **79.8** | 2845 | 55.0 | 3302 | 64.4 | 3439 |
> | 750 | 85.6 | 1942 | 36.7 | 5505 | 78.0 | 3037 | 55.4 | 3365 | 64.1 | 3548 |
> | 1000 | **86.2** | 2012 | 36.7 | 6158 | 79.0 | 3165 | **55.9** | 3552 | **64.5** | 3722 |
>
> $d$ is used to control the updated length budget. When it is set to a small value, e.g. 250, LLM will suffer obvious performance drop, e.g., 61.9 avg. accuracy. To explain, when $d$ is too small, ABPO will allocate insufficient budgets to the newly income hard examples, also leading to inferior optimization. However, when given proper budgets, e.g., 500, the training of LLM is greatly improved, also suggesting the robustness of our ABPO.
>
> **Overall, the design of ABPO makes it robust to these hyper-parameters in most cases. And the exceptions are akin to the motivation of ABPO and well tackled by the paper.**
>
> **Comment 4:**
> The approach does not fundamentally modify GRPO—it primarily wraps a training schedule around it. It seems to be ineffective if we do not revisit the same question again?
>
> **Response:**  Thanks for your insightful comment. In this paper, **we do apply ABPO to iterate through the training examples for multiple times, which is a common setting for mathematical tasks and the same as compared methods** [7,8,9].  **We also understand your concerns about the one-shot tuning on large-scale data. However, we think that our ABPO is also applicable to your mentioned cases.**
>
> In terms of your mentioned one-shot training case, **our ABPO can shift from the example-wise to task-wise RL curriculum learning.** Simply put, ABPO will consider the task-wise difficulty to schedule the training and token budget. Concretely, advance LLMs often collect examples for very granular tasks, e.g., GPT or Qwen series, thus ABPO can dynamically allocate the budget and update the curriculum learning steps at task level, where the three example pools are also constructed based on different granular tasks. In this case, **we can assess the task difficulties via sampling their examples rather than evaluating all examples.** In this way, ABPO is likely to achieve the balance between training efficiency and model performance.
>
> Thanks again for your suggestions, and we will leave this topic in our near-future works.

---

> ### Author Response · Authors · 2025-11-21
>
> **Comment 5:**
> How sensitive is ABPO to the choice of τ₀, τ₁, and d? Could these hyperparameters be learned or adapted automatically?
>
>
> **Response:** Thanks for your comment. The robustness of our ABPO has been shown in the previous tables and discussed above.
>
> **In terms of your mentioned dynamic (automatic) hyper-parameter setting, we think that this strategy is feasible to some extend, but may not be the optimal one.**
> For example, the threshold $\tau_0$ used to define the mastered and learning examples can indeed be dynamically set by averaging the accuracies of the examples with more correct CoT answers, e.g., 3/4 CoTs in a group are correct.  Even so, **this setting still relies on empirical knowledge to define what types of examples should be evaluated, and may be more unstable than our existing settings for ABPO.**
>
> Overall, we appreciate your in-depth suggestions, and will add the above discussions to our new submission.
>
>
> **Comment 6:** How does ABPO behave when the pool thresholds are poorly chosen—does training diverge or stagnate?
>
> **Response:** Thanks for your constructive comments. We report the ablations of $\tau_0$ and $\tau_1$ in the following tables (the default settings are marked with †).
>
> | τ₀ | MATH500 Acc | Len | AIME Acc | Len | AMC Acc | Len | Olympiad Acc | Len | Avg Acc | Len |
> |----|-------------|-----|-----------|------|----------|------|--------------|------|-----------|------|
> | 0.10 | 86.0 | **1813** | 32.7 | **5645** | 78.3 | **2778** | 54.7 | **3245** | 62.9 | **3370** |
> | **0.20†** | 85.6 | 1874 | **37.3** | 5736 | 79.8 | 2845 | 55.0 | 3302 | **64.4** | 3439 |
> | 0.30 | 85.6 | 1893 | 35.7 | 5697 | **80.5** | 3096 | **55.7** | 3303 | 64.4 | 3497 |
> | 0.40 | **87.0** | 1945 | 33.3 | 6109 |79.2 | 3078 | 55.0 | 3459 | 63.7 | 3634 |
>
> | τ₁ | MATH500 Acc | Len | AIME Acc | Len | AMC Acc | Len | Olympiad Acc | Len | Avg Acc | Len |
> |----|-------------|-----|-----------|------|----------|------|--------------|------|-----------|------|
> | 0.90 | **86.8** | **1837** | 33.3 | **5587** | 78.0 | 3133 | **55.5** | **3295** | 63.4 | 3463 |
> | **0.85†** | 85.6 | 1874 | **37.3** | 5736 | **79.8** | **2845** | 55.0 | 3302 | **64.4** | **3439** |
> | 0.80 | 86.3 | 1898 | 35.0 | 5628 | 77.3 | 2949 | 55.2 | 3304 | 63.4 | 3474 |
> | 0.75 | 86.3 | 1899 | 35.0 | 5704 | 77.0 | 3048 | **55.5** | 3393 | 63.5 | 3511 |
>
> As discussed above, $\tau_1$ is the threshold for updating the mastered examples, which is less sensitive to its settings.
> $\tau_0$ is used for putting the hard examples to the learning pools, which is robust at most cases.  The only exception is the setting of $\tau_0$ for the most difficult task AIME.
>
> This exception can well respond to your mentioned training diverge or stagnate.
> As discussed in the previous comments, **when $\tau_0$ is set by a small value like 0.1, ABPO will put more overly difficult examples to the learning pools, which instead will disrupt the learning of LLM under the curriculum learning principle.** In other words, these examples are too difficult to learn for LLM with the low token budgets, making their optimization less effective.
>
> **When $\tau_0$ is set by a large value, e.g., 0.4, the learning of difficult examples will stagnate to some extend.** Specifically, a larger value of $\tau_0$ means that the LLM needs to obtain higher avg. accuracy of the learning examples. To this end, these difficult examples will be more often kept in the learning pool, which also makes the examples of the hard pools difficult to be incorporated into the learning pool.
>
> Overall, ABPO is insensitive to most settings, and the only exception is the poorly setting of $\tau_0$, which may shows training diverge or stagnate on the most difficult examples.

---

> ### Author Response · Authors · 2025-11-21
>
> **Comment 7:** I recently came across this paper Optimizing Anytime Reasoning via Budget Relative Policy Optimization [10] which shares very similar motivation and method. Instead of dynamically setting budget, this paper uses the dense reward framework to let the model learn the optimal budget by itself. It would be nice if you can discuss your differences and how your method is fundamentally better than the baseline.
>
> **Response:** Thanks for recommending this excellent work.  After carefully reading this paper, we think that our ABPO is greatly different to BRPO in terms of motivation, principle and implementation:
>
> **(1) Difference in Objectives and Setup.** BRPO targets at the efficient CoT optimization under a *fixed global budget range*. It optimizes the CoT efficiency within a predetermined budget, which is accomplished by using a dense reward function to compare relative accuracy gains between different fixed budgets. In other words, BRPO improves policy quality *given* a fixed budget allocation.
>
> In contrast, ABPO is motivated by achieving *sample-wise adaptive budgeting*. Instead of assuming that the budget range is fixed in advance, ABPO explicitly encourages the model to generate shorter CoTs for easy examples and longer CoTs for harder ones, which is supported by the curriculum-learning based dynamic scheduling designs.
>
> **(2) Difference in Reward Functions.** BRPO adopts a dense budget-relative reward which aggregates reward contributions across multiple fixed budget levels. This dense reward formulation provides continuous feedback over a set of fixed token budgets, allowing the policy to be trained to favor length-performance trade-offs across different budget levels.
>
> In contrast, ABPO uses a sparse reward that is computed under a *difficulty-aware, dynamically allocated budget*, rather than comparing across multiple fixed budgets. Specifically, for each training instance, the model receives a binary reward indicating whether the generated solution is correct under its *current* token budget, which is adaptively assigned based on the sample's difficulty level. This design encourages efficient and targeted learning by aligning reward signals with curriculum-informed budget schedules.
>
> **(3) Performance Comparison.** Since BRPO does not release model checkpoints and the detailed accuracy–length tables, we reference its performance plot charts in Fig. 4 for a quick comparison:
>
> - On MATH500, ABPO (R1-Distill-1.5B) reaches 85.2 accuracy with only 1434 average tokens, while BRPO appears to require around 6000 tokens for similar performance.
> - On AIME, ABPO achieves 28.7 accuracy with 3223 tokens, whereas BRPO needs roughly 5000 tokens to match this level.
>
>
> Overall, we think that our method and BRPO are two effective methods following different principles and methodologies, of which contributions are orthogonal. We will follow your suggestion to cite and discuss this work in our paper.
>
> [1] Bengio, Yoshua, et al. "Curriculum learning." Proceedings of the 26th annual international conference on machine learning. 2009.
>
> [2] Alex Graves, Marc G. Bellemare, Jacob Menick, Rémi Munos, and Koray Kavukcuoglu. 2017. Automated curriculum learning for neural networks. In Proceedings of the 34th International Conference on Machine Learning - Volume 70 (ICML'17). JMLR.org, 1311–1320.
>
> [3] Hacohen, Guy, and Daphna Weinshall. "On the power of curriculum learning in training deep networks." International conference on machine learning. PMLR, 2019.
>
> [4] Xi, Zhiheng, et al. "Training large language models for reasoning through reverse curriculum reinforcement learning." arXiv preprint arXiv:2402.05808 (2024).
>
> [5] Shi, Taiwei, et al. "Efficient reinforcement finetuning via adaptive curriculum learning." arXiv preprint arXiv:2504.05520 (2025).
>
> [6] Shao, Zhihong, et al. "Deepseekmath: Pushing the limits of mathematical reasoning in open language models." arXiv preprint arXiv:2402.03300 (2024).
>
> [7] Huang, Chengyu, Zhengxin Zhang, and Claire Cardie. "HAPO: Training Language Models to Reason Concisely via History-Aware Policy Optimization." arXiv preprint arXiv:2505.11225 (2025).
>
> [8] Yi, Jingyang, Jiazheng Wang, and Sida Li. "Shorterbetter: Guiding reasoning models to find optimal inference length for efficient reasoning." arXiv preprint arXiv:2504.21370 (2025).
>
> [9] Hou, Bairu, et al. "Thinkprune: Pruning long chain-of-thought of llms via reinforcement learning." arXiv preprint arXiv:2504.01296 (2025).
>
> [10] Qi, Penghui, et al. "Optimizing anytime reasoning via budget relative policy optimization." arXiv preprint arXiv:2505.13438 (2025).

---

> > ### Author Response · Authors · 2025-11-25
> >
> > Dear Reviewer Yb4b:
> >
> > Thanks again for your great effort in reviewing our paper. In our rebuttal, we have carefully responded to all your concerns, such as the theoretical grounding of our ABPO, the formal justification of our curriculum structure, the ablations of key hyper-parameters, the behaviors of our ABPO training and the discussion with your suggested work.
> >
> > We hope that our rebuttal can well address your concerns. And we also look forward to hearing from you and will be very delighted to answer any further questions you may have.
> >
> > Best regards
> >
> > The authors.

---

### Official Review · Reviewer_q95z · 2025-11-03

**Soundness:** 2
**Presentation:** 2
**Contribution:** 2
**Rating:** 2
**Confidence:** 5

**Summary:**

The paper proposes ABPO, a reinforcement-learning framework that adapts the token budget per example to balance CoT efficiency (shorter traces) and accuracy. It builds on GRPO by (i) adding a budget-aware reward, and (ii) organizing data into dynamically updated mastered/learning/hard pools with curriculum-style scheduling and periodic review. Experiments across multiple math benchmarks and three base models show improved length–accuracy trade-offs relative to fixed-budget.

**Strengths:**

1. The paper is generally well-written and structured, followed by the motivation and explanation.
2. The separation of the token allocation decision from the generation process is a clean and novel architectural contribution.
3. The experimental results validate the central hypothesis: the model learns to assign smaller budgets to easier problems and reserves larger budgets for more difficult ones.

**Weaknesses:**

1. The reward design is based on a hard cutoff binary scheme, which does not distinguish between reasoning traces that are slightly longer than the target and those that are significantly longer. I suggest incorporating a soft budget penalty, similar to the L1-Max penalty, to encourage smoother control over reasoning length.
2. The paper introduces a number of hyperparameters( $\tau_0$, $\tau_1$, $\lambda$, $\alpha$, $d$, and $t_0$) without providing evidence of tuning.
3. The authors evaluate with $K = 3, 5, 10$ rollouts. However, it is common practice to perform 16 or 32 rollouts for a fair evaluation and to compute pass@1 scores. Using only 10 rollouts may not be sufficient.
4. The proposed method only outperforms the DeepSeek-1.5B model in Table 1, but performs worse than the base DeepSeek-7B and DeepScaleR-1.5B models.
5. In Table 2, the authors compare only against DeepSeek-1.5B, while DeepScaleR exhibits stronger baseline performance. I recommend evaluating the method based on DeepScaleR, and for fairness, performing fixed-budget comparisons for both ABPO and DeepScaleR—for example, by applying early stopping to DeepScaleR under equivalent token budgets.
6. In Figure 3, presenting only a single point is insufficient for fair comparison. To better demonstrate performance–efficiency trade-offs, the authors should ensure similar token usage between ABPO and L1-Max, and include results for multiple token budgets (e.g., 512, 1024, 2048 tokens).

**Questions:**

See Weaknesses

---

> ### Author Response · Authors · 2025-11-21
>
> ## Comment to Reviewer q95z
> Thanks a lot for your time and efforts in reviewing our paper as well as your encouraging comments on our work, such as *"the paper is generally well-written and structured"*, *"a clean and novel architectural contribution"* and *"the experimental results validate the central hypothesis"*. Below, we respond to your key concerns point by point.
>
> **Comment 1:** The reward design is based on a hard cutoff binary scheme, which does not distinguish between reasoning traces that are slightly longer than the target and those that are significantly longer. I suggest incorporating a soft budget penalty, similar to the L1-Max penalty, to encourage smoother control over reasoning length.
>
> **Response:** Thanks for your kindly suggestion.  **We supplement the following experiments of using L1-Max's reward design in our ABPO as you requested.** It can be seen that the reward design in **our ABPO is superior than the soft penalty of L1-Max, which can obtain 5.6% avg. performance improvement while only increasing 348 avg. length.** Notably, **on the most difficult task AIME, our performance gain can be up to 19.2%, and the avg. CoT length is only 5736, about 63% of baseline.**
>
> | Method                  | MATH500  | Len         | AIME     | Len         | AMC      | Len         | Olympiad  | Len         | Avg      | Len         |
> | ----------------------- | -------- | ----------- | -------- | ----------- | -------- | ----------- | -------- | ----------- | -------- | ----------- |
> | DeepScaler-1.5B-Preview | 88.2     | 3155        | 40.0     | 9041        | 81.3     | 5067        | 58.3     | 5718        | 66.9     | 5745        |
> | L1-Max 1.5B             | 84.5     | **1527**    | 27.0     | **2290**    | 75.5     | **1768**    | 51.1     | **1733**    | 59.5     | **1830**    |
> | ABPO w L1-Max Reward    | 84.5     | $\underline{1655}$ | 31.3     | $\underline{5039}$ | 74.3     | $\underline{2712}$ | 53.8     | $\underline{2959}$ | $\underline{61.0}$     | $\underline{3091}$ |
> | **ABPO(ours)**          | **85.6** | 1874        | **37.3** | 5736        | **79.8** | 2845        | **55.0** | 3302        | **64.4** | 3439        |
>
>
> In terms of these superior results, our analysis is as follow:
>
> (1)  **Our curriculum learning designs in ABPO well consider the difficulties of examples to LLM,** i.e., the construction and dynamic scheduling of the mastered, learning and hard pools, thus **the entire training process is in-fact not a hard-cut optimization but a smooth and progressive control over reasoning length,** as you suggested.
>
> (2) Under the curriculum learning scheme, **the learning examples are within a suitable range for the LLM's difficulty and the token budget,** e.g., about half of CoTs in a group are correct. In this case, it is more important to enhance the penalty signals for the incorrect CoTs, and **using the soft penalty design of L1-Max will instead make the rewards difficult to distinguish, thereby undermining the optimization.**
>
> (3) Moreover, compared to the default reward of our ABPO, **using the soft reward function of L1-Max will include inappropriated hard examples to the learning pools.**  To explain, the use of soft reward will obviously increase the avg. reward value of the learning examples. In this case, some hard examples may be added to the learning pool prematurely, which does not fully meet the principle of curriculum learning and impede the optimization instead.
>
> Overall, our ABPO uses the curriculum learning principle to build a smooth and progressive training process for the control of CoT length, and our hard reward design is more suitable in this training scheme.

---

> ### Author Response · Authors · 2025-11-21
>
> **Comment 2:** The paper introduces a number of hyperparameters $(\tau_0, \tau_1, \lambda, \alpha, d, t_0)$ without providing evidence of tuning.
>
>
> **Response:** Thanks for your suggestion. Before the submission, **we ever conducted some quick validations about the selection of these hyper-parameters.** Following your suggestion, we report their complete results in the following tables (the default settings are marked with †).
>
> | τ₀ | MATH500 Acc | Len | AIME Acc | Len | AMC Acc | Len | Olympiad Acc | Len | Avg Acc | Len |
> |----|-------------|-----|-----------|------|----------|------|--------------|------|-----------|------|
> | 0.10 | 86.0 | **1813** | 32.7 | **5645** | 78.3 | **2778** | 54.7 | **3245** | 62.9 | **3370** |
> | **0.20†** | 85.6 | 1874 | **37.3** | 5736 | 79.8 | 2845 | 55.0 | 3302 | **64.4** | 3439 |
> | 0.30 | 85.6 | 1893 | 35.7 | 5697 | **80.5** | 3096 | **55.7** | 3303 | 64.4 | 3497 |
> | 0.40 | **87.0** | 1945 | 33.3 | 6109 |79.2 | 3078 | 55.0 | 3459 | 63.7 | 3634 |
>
>
>
> | τ₁ | MATH500 Acc | Len | AIME Acc | Len | AMC Acc | Len | Olympiad Acc | Len | Avg Acc | Len |
> |----|-------------|-----|-----------|------|----------|------|--------------|------|-----------|------|
> | 0.90 | **86.8** | **1837** | 33.3 | **5587** | 78.0 | 3133 | **55.5** | **3295** | 63.4 | 3463 |
> | **0.85†** | 85.6 | 1874 | **37.3** | 5736 | **79.8** | **2845** | 55.0 | 3302 | **64.4** | **3439** |
> | 0.80 | 86.3 | 1898 | 35.0 | 5628 | 77.3 | 2949 | 55.2 | 3304 | 63.4 | 3474 |
> | 0.75 | 86.3 | 1899 | 35.0 | 5704 | 77.0 | 3048 | **55.5** | 3393 | 63.5 | 3511 |
>
>
> $\tau_0$ and $\tau_1$ are two thresholds used for moving the hard and moderate examples to the learning and mastered pools in ABPO, respectively.
> Form these results, **we can first see that ABPO is robust to these two hyper-parameters in most cases. The only exception is the most difficult benchmark, AIME, which is also the one where most existing methods perform poorly.**
> Specifically, $\tau_0$ is more sensitive to this benchmark than $\tau_1$. To explain, when $\tau_0$ is too small, *e.g.*, 0.1, ABPO will put more overly difficult examples  to the learning pools, which are beyond the learning range of LLM under the low token budget, a similar problem to the L1_Max reward discussed above. Similarly, when the threshold is set too high, LLM will consistently fail to achieve the expected average accuracy of $\tau_0$, making it unable to learn more difficult examples.
>
>
> | d | MATH500 Acc | Len | AIME Acc | Len | AMC Acc | Len | Olympiad Acc | Len | Avg Acc | Len |
> |----|-------------|------|-----------|------|----------|------|--------------|------|-----------|------|
> | 250 | 85.8 | **1785** | 29.7 | **5325** | 76.3 | **2747** | 55.8 | **3088** | 61.9 | **3286** |
> | **500†**| 85.6 | 1874 | **37.3** | 5736 | **79.8** | 2845 | 55.0 | 3302 | 64.4 | 3439 |
> | 750 | 85.6 | 1942 | 36.7 | 5505 | 78.0 | 3037 | 55.4 | 3365 | 64.1 | 3548 |
> | 1000 | **86.2** | 2012 | 36.7 | 6158 | 79.0 | 3165 | **55.9** | 3552 | **64.5** | 3722 |
>
> | t₀ | MATH500 Acc | Len | AIME Acc | Len | AMC Acc | Len | Olympiad Acc | Len | Avg Acc | Len |
> |----|-------------|------|-----------|------|----------|------|--------------|------|-----------|------|
> | 1500 | 85.9 | **1787** | 36.3 | **5351** | 78.8 | 2902 | 55.4 | **3199** | 64.1 | **3310** |
> | **2000†** | 85.6 | 1874 | **37.3** | 5736 | 79.8 | **2845** | 55.0 | 3302 | 64.4 | 3439 |
> | 2500 | **87.0** | 1966 | 36.0 | 5928 | **80.5** | 3062 | **56.1** | 3522 | **64.9** | 3620 |
>
> In terms of $d$ and $t_0$, they are related to the token length settings for the update of the learning pools. $t_0$ denotes the initial token budget. **ABPO is less effected by this hyper-parameter in terms of the avg. accuracy, but $t_0$ will slightly affect the final length of CoTs, but still marginal.**
>
> $d$ is used to control the updated length budget. When it is set to a small value, e.g. 250, LLM will suffer obvious performance drop, e.g., 61.9 avg. accuracy. To explain, when $d$ is too small, ABPO will allocate insufficient budgets to the newly income hard examples, also leading to inferior optimization. **However, when given proper budgets, e.g., 500, the training of LLM is greatly improved, also suggesting the robustness of our ABPO.**

---

> ### Author Response · Authors · 2025-11-21
>
> **Comment 2:** The paper introduces a number of hyperparameters $(\tau_0, \tau_1, \lambda, \alpha, d, t_0)$ without providing evidence of tuning.
>
> | λ | MATH500 Acc | Len | AIME Acc | Len | AMC Acc | Len | Olympiad Acc | Len | Avg Acc | Len |
> |----|-------------|------|-----------|------|----------|------|--------------|------|-----------|------|
> | 0.6 | 85.7 | 1857 | 35.7 | 5612 | 79.3 | 2914 | 55.9 | 3340 | 64.2 | 3431 |
> | **0.5†** | 85.6 | 1874 | **37.3** | 5736 | 79.8 | **2845** | 55.0 | 3302 | 64.4 | 3439 |
> | 0.4 | **86.2** | **1806** | 34.3 | **5538** | 80.8 | 2855 | 55.3 | 3238 | 64.2 | **3359** |
> | 0.3 | 86.1 | 1881 | 36.0 | 5588 | **81.5** | 3029 | **56.0** | **3232** | **64.9** | 3433 |
>
> | α | MATH500 Acc | Len | AIME Acc | Len | AMC Acc | Len | Olympiad Acc | Len | Avg Acc | Len |
> |----|-------------|------|-----------|------|----------|------|--------------|------|-----------|------|
> | **0.1†**| 85.6 | **1874** | **37.3** | 5736 | **79.8** | **2845** | 55.0 | **3302** | **64.4** | **3439** |
> | 0.2 | **86.9** | 1931 | 32.7 | 5718 | 79.3 | 2983 | **56.0** | 3363 | 63.7 | 3499 |
> | 0.3 | 86.3 | 1938 | 35.7 | **5625** | 78.3 | 2905 | 55.2 | 3337 | 63.9 | 3451 |
>
>
> $\lambda$ denotes the proportion of the learning examples, which also determine the update of both learning and hard pools. As can be seen, its effects to ABPO is limited. $\alpha$ denotes the review rate of the mastered examples, where a smaller value is more beneficial to the final training results, but it is still not very sensitive.
>
> Overall, **the careful designs of ABPO makes it robust to these hyper-parameters in most cases. And the exceptions are akin to the motivation of ABPO and have be well tackled by our previous quick validations.** We will follow your suggestion to add these ablations to our new submission.
>
>
> **Comment 3:** The authors evaluate with $(K = 3,5,10)$ rollouts. However, it is common practice to perform 16 or 32 rollouts for a fair evaluation and to compute pass@1 scores. Using only 10 rollouts may not be sufficient.
>
> **Response:** Thanks for your careful comment. In fact, **we follow previous work termed *Training Efficiently* [1] (NeurIPS 2025)** *e.g.*, which adopts \(K=3\) rollouts for MATH500 and \(K=10\) for AIME. **This setting is also close to the compared methods.** For instance, *HAPO* [2] uses \(K=4\) rollouts on MATH500, and *ShorterBetter* [3] (NeurIPS 2025) evaluates all datasets with \(K=1\). For more fair comparisons, we tested these methods under the same setting in our current submission, *i.e.*, $K=(3,5,10)$.
>
> Following your suggestion, **we further supplement the experiments using \(K=16\) rollouts on the DeepScaleR-1.5B model.** († denotes the settings of our paper.)
>
> | Method             | Evaluation        | MATH500 | Len  | AES  | AIME | Len  | AES    | AMC | Len  | AES  | Olympiad | Len  | AES  | Avg  | Len  | AES  |
> |-------------------|-------------------|---------|------|------|------|------|--------|-----|------|------|---------|------|------|------|------|------|
> | DeepScaleR-1.5B    | K=16              | 88.3    | 3148 | -    | 41.0 | 9200 | -      | 81.4| 5189 | -    | 58.6    | 5678 | -    | 67.3 | 5804 | -    |
> | DeepScaleR-1.5B    | K=(3,10,10,5)†   | 88.2    | 3155 | -    | 40.0 | 9041 | -      | 81.3| 5067 | -    | 58.3    | 5718 | -    | 66.9 | 5745 | -    |
> | L1-Max           | K=16              | 83.2    | 1331 | 0.29 | 23.1 | 2064 | -1.40  | 74.7| 1587 | 0.28 | 50.6    | 1545 | 0.04 | 57.9 | 1632 | 0.02 |
> | L1-Max         |  K=(3,10,10,5)†   | 84.5    | 1527 | 0.31 | 27.0 | 2290 | -0.88  | 75.5| 1768 | 0.30 | 51.1    | 1733 | 0.07 | 59.5 | 1830 | 0.13 |
> | Thinkprune         | K=16              | 86.5    | 2034 | 0.25 | 34.4 | 5720 | -0.43  | 79.5| 3078 | 0.29 | 56.1    | 3542 | 0.16 | 64.1 | 3594 | 0.14 |
> | Thinkprune         |  K=(3,10,10,5)†   | 85.5    | 2029 | 0.21 | 34.7 | 5743 | -0.30  | 78.5| 3125 | 0.21 | 55.0    | 3532 | 0.10 | 63.4 | 3607 | 0.11 |
> | ABPO         | K=16              | 85.6    | 1874 | 0.26 | 37.3 | 5736 | 0.03   | 79.8| 2845 | 0.35 | 55.0    | 3302 | 0.14 | 64.4 | 3439 | 0.21 |
> | ABPO               | K=(3,10,10,5)†   | 85.9    | 1872 | 0.27 | 36.0 | 5566 | -0.21  | 79.8| 2990 | 0.33 | 55.6    | 3308 | 0.16 | 64.3 | 3434 | 0.19 |
>
>
> Similar observations and conclusions can be also obtained from the new experimental results under your suggested settings. For instance, our ABPO still achieve the best avg. accuracy and AES compared to other efficient CoT methods. Besides, ABPO can still allocate proper token budgets for different tasks of varying difficulties, *e.g.*, 1874 for MATH500 and 3302 for Olympiad. Moreover, **the performance gaps of ABPO between two settings are also marginal,** *e.g.*, 64.4 v.s. 64.3 avg. accuracy and 3439 v.s. 3434 avg. length.
>
> Overall, these new results well confirm the designs of our ABPO as well as its robustness.

---

> ### Author Response · Authors · 2025-11-21
>
> **Comment 4:** The proposed method only outperforms the DeepSeek-1.5B model in Table 1, but performs worse than the base DeepSeek-7B and DeepScaleR-1.5B models.
>
> **Comment 5:** In Table 2, the authors compare only against DeepSeek-1.5B, while DeepScaleR exhibits stronger baseline performance. I recommend evaluating the method based on DeepScaleR, and for fairness, performing fixed-budget comparisons for both ABPO and DeepScaleR—for example, by applying early stopping to DeepScaleR under equivalent token budgets.
>
> **Response for Comment 4&5:** Thanks again for your detailed comments. We follow your suggestions to supplement **the new comparisons with DeepScaleR-1.5B and DeepSeek-7B under your suggested fixed-budget settings.**
>
> | Method            | Token | MATH500 Acc | Len  | AIME Acc | Len  | AMC Acc | Len  | Olympiad Acc | Len  | Avg  | Len  |
> |------------------|--------|-------------|------|----------|------|---------|------|-------------|------|------|------|
> | DeepScaleR-1.5B-Preview  | 2000   | 75.1        | 1697 | 6.3      | 2051 | 52.3    | 1928 | 39.2        | 1958 | 43.2 | 1909 |
> | **ABPO**         | 2000   | **80.7**    | **1326** | **16.7** | **2015** | **63.8** | **1717** | **46.5** | **1775** | **51.9** | **1708** |
> | DeepScaleR-1.5B-Preview  | 3000   | 80.5        | 2100 | 19.0     | 3028 | 63.0    | 2585 | 47.3        | 2686 | 52.4 | 2600 |
> | **ABPO**         | 3000   | **83.6**    | **1503** | **23.0** | **2806** | **72.5** | **2093** | **51.1** | **2241** | **57.5** | **2161** |
> | DeepScaleR-1.5B-Preview  | 4000   | 83.8        | 2431 | 24.0     | 3889 | 66.0    | 3071 | 48.3        | 3295 | 55.5 | 3172 |
> | **ABPO**         | 4000   | **84.3**    | **1603** | **31.0** | **3483** | **75.8** | **2349** | **52.6** | **2457** | **60.9** | **2473** |
> | DeepScaler-1.5B-Preview |32768 |  **88.2**     | 3155        | **40.0**     | 9041        | **81.3**     | 5067        | **58.3**     | 5718        | **66.9**     | 5745        |
> | **ABPO**          |32768 |  85.6 | 1874        | 37.3 | 5736        | 79.8 | **2845**        | 55.0 | **3302**        | 64.4 | **3439**        |
>
>
>
> From these results, we can first see that DeepScaleR-1.5B encounters obvious performance drops on all benchmarks and metrics under the fix-budged settings.
> To explain, **the superior performance of DeepScaleR-1.5B stems from its GRPO training with excessive expenditures,** *e.g.*, using up to 24$k$ token budgets[4]. In this case, DeepScaleR-1.5B can well accommodate different math tasks via its RL training, and also exhibits superior capabilities than the only SFT-tuned DeepSeek-R1-1.5B.
> However, **without careful efficiency designs,  DeepScaleR-1.5B is hard to meet the requirements of fixed-budget settings.**
>
> In stark contrast, **our ABPO can well handle the fixed-budget requirements, although it is trained with a maximum budget of 5k tokens.** For instance, under the setting of 2k budget, ABPO can outperform DeepSeekR-1.5B by 20% avg. accuracy, while using even fewer tokens. Similarly, ABPO is also consistently and obviously better than DeepSeekR-1.5B under the 3k and 4k budgets in terms of both accuracy and actual token length. Similar cases can be also seen on the comparisons to DeepSeek-Distill-Qwen-7B.
>
> | Method              | Token | MATH500 Acc | Len      | AIME Acc | Len      | AMC Acc  | Len      | Olympiad Acc | Len      | Avg      | Len      |
> | ---------------------------- | ----- | ----------- | -------- | -------- | -------- | -------- | -------- | ------------ | -------- | -------- | -------- |
> | DeepSeek-Distill-Qwen-7B     | 2000  | 81.5        | 1782     | 17.3     | 2036     | 55.3     | 1952     | 46.8         | 1979     | 50.2     | 1937     |
> | **ABPO**                     | 2000  | **83.7**    | **1410** | **21.3** | **2016** | **67.8** | **1790** | **51.8**     | **1819** | **56.2** | **1759** |
> | DeepSeek-Distill-Qwen-7B     | 3000  | 85.4        | 2292     | 26.7     | 3019     | 69.3     | 2688     | 54.3         | 2784     | 58.9     | 2696     |
> | **ABPO**                     | 3000  | **86.3**    | **1648** | **32.0** | **2875** | **77.0** | **2256** | **57.3**     | **2371** | **63.2** | **2288** |
> | DeepSeek-Distill-Qwen-7B     | 4000  | 88.3        | 2604     | 30.0     | 3938     | 77.0     | 3278     | 56.6         | 3455     | 63.0     | 3319     |
> | **ABPO**                     | 4000  | **90.1**    | **1782** | **37.3** | **3605** | **81.0** | **2561** | **59.3**     | **2802** | **66.9** | **2688** |
> | DeepSeek-Distill-Qwen-7B | 32768 | **93.2**    | 4086     | **56.3** | 12837    | **89.0** | 6484     | **66.1**     | 8686     | **76.2** | 8023     |
> | **ABPO**                         | 32768 | 92.2        | **2243** | 48.7     | **8493** | 87.3     | **3511** | 63.5         | **4852** | 72.9     | **4775** |
>
> These results well validate the effectiveness of our ABPO towards efficient adaptive CoT reasoning, and we will follow your suggestions to add them to our new submission.

---

> ### Author Response · Authors · 2025-11-21
>
> **Comment 6:** In Figure 3, presenting only a single point is insufficient for fair comparison. To better demonstrate performance–efficiency trade-offs, the authors should ensure similar token usage between ABPO and L1-Max, and include results for multiple token budgets (e.g., 512, 1024, 2048 tokens).
>
>
> **Response:** Thanks for your suggestion. **We have supplemented experiments comparing ABPO, L1-Max, and DeepScaleR-1.5B-Preview under multiple token budgets.**
>
>
> | Method                     | Token Budget | Acc   | Actual Len | Token Budget | Acc   | Actual Len | Token Budget | Acc   | Actual Len |
> |----------------------------|--------------|-------|------------|--------------|-------|------------|--------------|-------|------------|
> | DeepScaler-1.5B-Preview    | 2000         | 43.2  | 1908       | 3000         | 52.4  | 2600       | 4000         | 55.5  | 3172       |
> | L1-Max                     | 2000         | **56.1** | **1286**   | 3000         | **58.8** | **1828**   | 4000         | 59.2  | **1622**   |
> | **ABPO (Ours)**                       | 2000         | 51.9  | 1708       | 3000         | 57.5  | 2160       | 4000         | **60.9** | 2473       |
>
> | Method                     | Token Budget | Acc   | Actual Len | Token Budget | Acc   | Actual Len | Token Budget | Acc   | Actual Len |
> |----------------------------|--------------|-------|------------|--------------|-------|------------|--------------|-------|------------|
> | DeepScaler-1.5B-Preview    | 5000         | 60.8  | 3619       | 6000         | 61.1  | 4031       | 32768        | **66.9**  | 5745       |
> | L1-Max                     | 5000         | 59.5  | **2049**   | 6000         | 59.0  | **2010**   | 32768        | 59.5  | **1829**   |
> | **ABPO (Ours)**                      | 5000         | **61.9** | 2708       | 6000         | **62.4** | 2900       | 32768        | 64.4  | 3439       |
>
>
>
> From the results, we can first see that **L1-Max performs  best under the low-budget settings, e.g., 2k-3k.** To explain, L1-Max is trained with a prompt that controls output length without distinguishing between question difficulty, which selects lengths uniformly between 100 and 4000. As a result, it tends to answer under lower budgets, which is more close to its training objective.
>
> However, as the token budget increases, the advantages of our ABPO become more obvious compared to L1-Max, e.g., 4k-6k. **Moreover, the performance gains of L1-Max are become very limited after the 4k budget. In contrast, our ABPO can be consistently improved.**
> In terms of DeepScaleR-1.5B, its capabilities are greatly constrained by fixed-budget settings, as discussed above.
>
> Overall, these results show that our ABPO can well tackle the math tasks of varying difficulties under different token budgets, which also further confirm our motivation and designs for adaptive-budgeting based CoT training.
>
> [1] Arora, Daman, and Andrea Zanette. "Training language models to reason efficiently." arXiv preprint arXiv:2502.04463 (2025).
>
> [2] Huang, Chengyu, Zhengxin Zhang, and Claire Cardie. "HAPO: Training Language Models to Reason Concisely via History-Aware Policy Optimization." arXiv preprint arXiv:2505.11225 (2025).
>
> [3] Yi, Jingyang, Jiazheng Wang, and Sida Li. "Shorterbetter: Guiding reasoning models to find optimal inference length for efficient reasoning." arXiv preprint arXiv:2504.21370 (2025).
>
> [4] Luo, Michael, et al. "Deepscaler: Surpassing o1-preview with a 1.5 b model by scaling rl, 2025." Notion Blog 3.4: 5.

---

> > ### Author Response · Authors · 2025-11-25
> >
> > Dear Reviewer q95z:
> >
> > Thanks again for your great effort in reviewing our paper.
> >
> > In our rebuttal, we have responded all of your concerns in detail, such as the use of soft budget penalty, the selection of hyper-parameters and rollouts, and more comparisons with DeepScaleR.
> >
> > We hope that our rebuttal can well address your concerns. And we also look forward to hearing from you and will be very delighted to answer any further questions you may have.
> >
> > Best regards
> >
> > The authors.

---

### Note · Authors · 2025-12-02

I have read and agree with the venue's withdrawal policy on behalf of myself and my co-authors.